

# Comparing proxy and model estimates of hydroclimate variability and change over the Common Era

*Hydro2k Consortium*: Jason E. Smerdon[1], Jürg Luterbacher[2,3], Steven J. Phipps[4], Kevin J. Anchukaitis[5], Toby Ault[6], Sloan Coats[7,8], Kim M. Cobb[9], Benjamin I. Cook[1,10], Chris Colose[10], Thomas Felis[11], Ailie Gallant[12], Johann H. Jungclaus[13], Bronwen Konecky[8], Allegra LeGrande[10], Sophie Lewis[14], Alex S. Lopatka[15], Wenmin Man[16], Justin S. Mankin[1,10], Justin T. Maxwell[17], Bette L. Otto-Bliesner[7], Judson W. Partin[18], Deepti Singh[1], Nathan J. Steiger[1], Samantha Stevenson[7], Jessica E. Tierney[19], Davide Zanchettin[20], Huan Zhang[2], Alyssa R. Atwood[9,21], Laia Andreu-Hayles[1], Seung H. Baek[1], Brendan Buckley[1], Edward R. Cook[1], Rosanne D'Arrigo[1], Sylvia G. Dee[22], Michael Griffiths[23], Charuta Kulkarni[24], Yochanan Kushnir[1], Flavio Lehner[7], Caroline Leland[1], Hans W. Linderholm[25], Atsushi Okazaki[26], Jonathan Palmer[27], Eduardo Piovano[28], Christoph C. Raible[29], Mukund P. Rao[1], Jacob Scheff[1], Gavin A. Schmidt[10], Richard Seager[1], Martin Widmann[31], A. Park Williams[1], Elena Xoplaki[2]

**Affiliations**

[1]Lamont-Doherty Earth Observatory of Columbia University, Palisades, NY, USA

[2]Department of Geography, Climatology, Climate Dynamics and Climate Change, Justus Liebig University of Giessen, Giessen, Germany

[3]Centre for International Development and Environmental Research, Justus Liebig University Giessen, Giessen, Germany

[4]Institute for Marine and Antarctic Studies, University of Tasmania, Hobart, Tasmania, Australia

[5]School of Geography and Development and Laboratory of Tree Ring Research, University of Arizona, Tucson, AZ, USA

[6]Department of Earth and Atmospheric Sciences, Cornell University, Ithaca, NY, USA

[7]National Center for Atmospheric Research, Boulder, CO, USA

[8]Cooperative Institute for Research in Environmental Sciences, University of Colorado, Boulder, CO, USA

[9]Department of Earth and Atmospheric Sciences, Georgia Institute of Technology, Atlanta, GA, USA

[10]NASA Goddard Institute for Space Studies, New York, NY, USA

[11]MARUM – Center for Marine Environmental Sciences, University of Bremen, Bremen, Germany

[12]School of Earth, Atmosphere and Environment, Monash University, Melbourne, Australia

[13]Max-Planck-Institute for Meteorology, Hamburg, Germany

[14]Fenner School of Environment and Society, The Australian National University, Canberra, ACT, Australia

[15]Department of Geology, University of Maryland, College Park, MA, USA

[16]LASG, Institute of Atmospheric Physics, Chinese Academy of Sciences, Beijing, China

[17]Department of Geography, Indiana University, Bloomington, IN, USA

[18]Institute for Geophysics, Jackson School of Geosciences, University of Texas at Austin, Austin, TX, USA

[19]Department of Geosciences, University of Arizona, Tucson, AZ, USA

[20]Department of Environmental Sciences, Informatics and Statistics, Ca' Foscari University of Venice, Venice, Italy

[21]Department of Geography, University of California, Berkeley, Berkeley, CA, USA

[22]Department of Earth, Environmental, and Planetary Sciences, Brown University, Providence, RI, USA





[23]Department of Environmental Science, William Paterson University, Wayne, NJ, USA

[24]Department of Earth and Environmental Sciences, The Graduate Center, City University of New York, New York, NY, USA

[25]Regional Climate Group, Department of Earth Sciences, University of Gothenburg, Gothenburg, Sweden

[26]Atmosphere and Ocean Research Institute, University of Tokyo, Tokyo, Japan

[27]Palaeontology, Geobiology and Earth Archives Research Centre, School of Biological, Earth and Environmental Sciences, University of New South Wales, Australia.

[28]Centro de Investigaciones en Ciencias de la Tierra (CICTERRA / CONICET - Universidad Nacional de Córdoba), Córdoba, Argentina

[29]Climate and Environmental Physics, Physics Institute and Oeschger Centre for Climate Change Research, University of Bern, Bern, Switzerland

[30]School of Geography, Earth and Environmental Sciences, University of Birmingham, Birmingham, UK

*Correspondence to*: Jason E. Smerdon (jsmerdon@ldeo.columbia.edu)




**Abstract.**

Water availability is fundamental to societies and ecosystems, but our understanding of variations in hydroclimate (including extreme events, flooding, and decadal periods of drought) is limited because of a paucity of modern instrumental observations that are distributed unevenly across the globe and only span parts of the 20th and 21st centuries. Such data coverage is insufficient for characterizing hydroclimate and its associated dynamics because of its multidecadal-to-centennial variability and highly regionalized spatial signature. High-resolution (seasonal to decadal) hydroclimatic proxies that span all or parts of the

Common Era (CE) and paleoclimate model simulations are therefore important tools for augmenting our understanding of hydroclimate variability. In particular, the comparison of the two sources of information is critical for addressing the uncertainties and limitations of both, while enriching each of their interpretations. We review the principal proxy data available for hydroclimatic reconstructions over the CE and highlight contemporary understanding of how these proxies are interpreted as hydroclimate indicators. We also review the available last-millennium simulations from fully-coupled climate models and

discuss several outstanding challenges associated with simulating hydroclimate variability and change over the CE. A specific review of simulated hydroclimatic changes forced by volcanic events is provided, as well as a discussion of expected improvements in estimated forcings, models and their implementation in the future. Our review of hydroclimatic proxies and last-millennium model simulations is used as the basis for articulating a variety of considerations and best practices for how to perform proxy-model comparisons of CE hydroclimate. This discussion provides a framework for how best to evaluate

hydroclimate variability and its associated dynamics using these comparisons, as well as how they can better inform interpretations of both proxy data and model simulations. We subsequently explore means of using proxy-model comparisons to better constrain and characterize future hydroclimate risks. This is explored specifically in the context of several examples that demonstrate how proxy-model comparisons can be used to quantitatively constrain future hydroclimatic risks as estimated from climate model projections.





**1. Introduction**

Research on the climate of the Common Era (CE; the last 2016 years) has exploded in scope and extent over the last two decades (e.g. Jones et al., 2009; Masson-Delmotte et al., 2013; PAGES 2k Consortium, 2013; Schmidt et al., 2014; Luterbacher et al., 2016; Smerdon and Pollack, 2016; Christiansen and Ljungqvist, 2017). Multiple developments have driven this expansion:

1)   improved interpretations of high-resolution proxies (e.g. tree rings, corals, lake sediments, documentary evidence, and speleothems; Masson-Delmotte et al., 2013);

2)   the development of proxy and multi-proxy networks to extend over the full CE and into poorly sampled regions such as the oceans, tropics and the Southern Hemisphere (e.g. Cook et al., 2010a; PAGES 2k Consortium, 2013; Neukom et al., 2014; Piovano et al., 2014; Cook et al., 2015b; McGregor et al., 2015; Palmer et al., 2015; Tierney et al., 2015a; Gergis

85       et al., 2016; Luterbacher et al., 2016);

3)   improved statistical methods for synthesizing proxy networks into index or gridded climate reconstructions that span all or parts of the globe (e.g. Smerdon, 2012; Tingley et al., 2012; Anchukaitis and Tierney, 2013; Dannenberg and Wise, 2013; Emile-Geay et al., 2013b, a; Steiger et al., 2014; Guillot et al., 2015; Hakim et al., 2016; Smerdon and Pollack, 2016);

4)   enhanced understanding of the forcings that have operated on the climate system during the CE (e.g. Schmidt et al., 2012; Sigl et al., 2015; Jungclaus et al., 2016);

5)   increased numbers of forced last-millennium simulations and simulation ensembles from fully-coupled climate models (e.g. Jungclaus et al., 2010; Fernandez-Donado et al., 2013; Schmidt et al., 2014; Jungclaus et al., 2016; Otto-Bliesner et al., 2016);

6)   comparisons between climate reconstructions and climate model simulations of the CE (e.g. Mann et al., 2009; Anchukaitis et al., 2010; Goosse et al., 2012b; Schmidt et al., 2014; Coats et al., 2015a; 2015b; Cook et al., 2015a; Neukom et al., 2015; PAGES 2k-PMIP3 Group, 2015; Luterbacher et al., 2016);

7)   the development of new tools and techniques for proxy-model comparisons, in particular proxy system models (PSMs; Evans et al., 2013) and data assimilation techniques (Goosse et al., 2010; Widmann et al., 2010; Goosse et al., 2012b;

100       Steiger et al., 2014; Hakim et al., 2016).

These efforts are being combined to describe how and why climate varied over the CE, with critical implications for how we model climate and anticipate the risks associated with anthropogenic climate change in the 21st century.

An important advance underlying several of the developments listed above was the inclusion of the last-millennium (*past1000*; 850-1850 CE) experiment into the Coupled Model Intercomparison Project Phase 5 (CMIP5) and Paleoclimate

Modelling Intercomparison Project Phase 3 (PMIP3) experimental protocol (e.g. Schmidt et al., 2011; Masson-Delmotte et al., 2013; Schmidt et al., 2014). For most of the last decade and a half, modeling CE climate relied primarily on computationally efficient energy balance models or Earth system models of intermediate complexity (e.g. Bertrand et al., 2002; Bauer et al., 2003; Crowley et al., 2003; Gerber et al., 2003; Goosse et al., 2010); only a few fully-coupled climate models were initially used to derive forced transient simulations of CE climate (González-Rouco et al., 2003; 2006; Ammann et al., 2007; Hofer et al.,

2011) or control simulations spanning several millennia (Raible et al., 2006; Wittenberg, 2009). More recently, however, and aided by the inclusion of the last-millennium simulation in the PMIP3 protocol, a large and growing ensemble of last-millennium simulations is available (Masson-Delmotte et al., 2013). Critically, the last-millennium simulations associated with PMIP3 were performed with a constrained set of boundary conditions and the same model resolutions and configurations as those completed for the historical and future projection experiments in CMIP5. This latter characteristic makes evaluations of



the last-millennium simulations directly comparable to the CMIP5 simulations (a similar convention will be used for
        PMIP4/CMIP6 (Jungclaus et al., 2016)) and therefore offers the opportunity to use proxy-model comparisons over the CE as
        truly quantitative constraints on future projections (e.g. Schmidt et al., 2014).

              Relative to comparisons over other paleoclimatic time periods that have a longer history of research, proxy-model
        comparisons specifically over the CE are a new endeavor and involve unique considerations because of the seasonal-to-annual
resolution of many CE proxies and the spatially resolved reconstructions that are possible because of their broad spatial
        sampling (e.g. Pinot et al., 1999; Braconnot et al., 2007; Otto-Bliesner et al., 2007; Braconnot et al., 2012; Schmidt et al., 2014).
        Initial efforts to compare proxies and models over the CE largely focused on temperature (e.g. Mann et al., 2009; Widmann et
        al., 2010; Goosse et al., 2012a; Fernandez-Donado et al., 2013; Masson-Delmotte et al., 2013; Phipps et al., 2013; Lehner et al.,
        2015; PAGES 2k-PMIP3 Group, 2015; Luterbacher et al., 2016), but more recent efforts have considered aspects of
hydroclimate (e.g. Anchukaitis et al., 2010; Coats et al., 2013b; Landrum et al., 2013; Tierney et al., 2013; Coats et al., 2015a;
        Coats et al., 2015b; Coats et al., 2015c; Gómez-Navarro et al., 2015; Smerdon et al., 2015; Ljungqvist et al., 2016; Otto-
        Bliesner et al., 2016; Stevenson et al., 2016). These hydroclimate proxy-model comparisons have been made possible not only
        by the growing ensemble of last-millennium simulations, but also by the expanding number of hydroclimate proxies and their
        syntheses into large-scale gridded hydroclimate reconstructions (e.g. Cook et al., 2004; Jones et al., 2009; Cook et al., 2010a;
Masson-Delmotte et al., 2013; Cook et al., 2015b; Palmer et al., 2015).

              Many of the same technical details associated with proxy-model comparisons of temperature (PAGES 2k-PMIP3
        Group, 2015) are also applicable to hydroclimate. The latter nevertheless requires unique considerations before meaningful
        comparisons can be made, many of which are being articulated by contemporary research. This review is therefore motivated
        by the emerging efforts to compare hydroclimate variability and change over the CE in proxies and models in order to better
understand these climate characteristics over decades to centuries, to jointly evaluate proxies and models over these timescales,
        and to ultimately constrain projections of risk in the 21[st] century associated with increasing anthropogenic emissions of
        greenhouse gases. We review the principal proxy data available for hydroclimatic reconstructions over the CE and highlight
        contemporary understanding of how these proxies are interpreted as hydroclimate indicators in Section 2. In Section 3, we
        review available last-millennium simulations and discuss several outstanding challenges associated with simulating
hydroclimate variability and change over the CE using fully-coupled climate models. A variety of considerations and best
        practices related to proxy-model comparisons of CE hydroclimate are discussed in Section 4, and Section 5 explores the
        possibility of using these comparisons to better constrain and characterize future hydroclimate risks. Each of these sections is
        summarized in Section 6, along with the important recommendations of our study.

**2. High-Resolution Hydroclimate Archives for the Common Era**
        The following subsections review the principal proxies available for hydroclimatic studies. We survey recent advances in the
        development and characterization of the proxy systems, as well as contemporary understanding of the uncertainties associated
        with their interpretation.

*2.1 Corals*
        Shallow-water corals of the tropical and subtropical oceans provide an archive of near-sea surface environmental conditions.
        Isotopic and elemental tracers incorporated into their massive carbonate skeletons during growth enable the construction of
        temperature and salinity-related proxy records (e.g. Lough, 2010). When analyzed at subseasonal resolution to reveal annual
        cycles, and supported by distinct annual density bands, corals can provide accurately dated near-monthly resolved





reconstructions of seasonal, interannual and decadal climate variability (e.g. Cobb et al., 2001; Linsley et al., 2004; Felis et al., 2009a; DeLong et al., 2012). Growing for up to a few centuries only, corals are particularly important in documenting ocean-atmosphere variability and change during the observational period back into the Little Ice Age. On rare occasions, modern and young fossil corals can be spliced together when supported by radiometric dating, which provides proxy records that cover time intervals spanning the last millennium (Cobb et al., 2003).

The most commonly analysed variable in coral skeletons is $\delta^{18}O$, which reflects a combination of both the temperature and the $\delta^{18}O$ of the seawater near the ocean surface. Seawater $\delta^{18}O$ ($\delta^{18}O_{sw}$) can be influenced by atmospheric processes such as the precipitation-evaporation (P-E) balance at the sea surface and the isotopic signature of precipitation, as well as oceanic processes such as the advection of surface currents and the upwelling of deeper water masses (Fairbanks et al., 1997; Conroy et al., 2014). In general, $\delta^{18}O_{sw}$ is highly correlated with ocean salinity, as they are both governed by the same fundamental

processes, but surface $\delta^{18}O_{sw}$-salinity slopes vary appreciably through space and time where they have been studied (LeGrande and Schmidt, 2011; Conroy et al., 2014). Importantly, the relative contributions of temperature and $\delta^{18}O_{sw}$ to the coral $\delta^{18}O$ signal may vary through time, as the ratio of temperature versus $\delta^{18}O_{sw}$ in the ocean likely varies on different time scales. Calibration efforts that rely on *in situ* observations of temperature and $\delta^{18}O_{sw}$ contributions to coral $\delta^{18}O$ over a period of several years therefore cannot be applied to constrain decadal to centennial-scale changes in ocean temperature from coral $\delta^{18}O$. Tools

such as the isotope-equipped Regional Ocean Modeling System (isoROMS; Stevenson et al., 2015a) hold great promise in probing the spatiotemporal variability of $\delta^{18}O_{sw}$, and its dynamical drivers, and remains under active development.

The coral Sr/Ca temperature proxy has the potential to decouple the temperature and $\delta^{18}O_{sw}$ signals from coral $\delta^{18}O$ records in order to deliver reconstructions of $\delta^{18}O_{sw}$ – a hydrologically-relevant climate variable – but only a handful of centuries-long coral Sr/Ca records currently exist (e.g. Linsley et al., 2000; Hendy et al., 2002; Linsley et al., 2004; Felis et al.,

2009a; Nurhati et al., 2011b; DeLong et al., 2012). While many studies suggest a high degree of reproducibility of coral Sr/Ca variations for corals of the most commonly used genus *Porites* (Stephans et al., 2004; Nurhati et al., 2009; DeLong et al., 2013), some studies find poor reproducibility of the Sr/Ca sea surface temperature (SST) proxy among contemporaneous *Porites* colonies recovered from the same reef (Linsley et al., 2006; Alpert et al., 2016). Other temperature proxies in corals include U/Ca (e.g. Hendy et al., 2002; Felis et al., 2009a), Li/Ca (e.g. Hathorne et al., 2013), and Sr-U (DeCarlo et al., 2016).

When paired with coral $\delta^{18}O$ measurements, the measurement of temperature-sensitive proxies such as coral Sr/Ca provide a tool for reconstructing $\delta^{18}O_{sw}$ changes of the surface ocean on seasonal, interannual and decadal time scales (e.g. Gagan et al., 2000; Hendy et al., 2002; Ren et al., 2003; Cahyarini et al., 2008; Felis et al., 2009a; Nurhati et al., 2011b). The calculation of paleo-$\delta^{18}O_{sw}$ changes, however, requires careful assessment of the associated errors (e.g. Schmidt, 1999; Cahyarini et al., 2008; Nurhati et al., 2011a; Giry et al., 2013), including quantitative estimates of the analytical error of both

coral $\delta^{18}O$ and the paleo-temperature proxy in question (i.e. coral Sr/Ca), as well as the uncertainties associated with the coral temperature proxy-SST relationship that are typically estimated via *in situ* calibration.

Early approaches used a sparse network of coral $\delta^{18}O$ records that were available at the time (11 sites) to reconstruct Pacific SST fields and the associated leading modes of large-scale variability over the last several centuries (Evans et al., 2002). Subsequently, the growing number of coral records has contributed to multiproxy reconstructions of tropical temperature

(D'Arrigo et al., 2009) and central equatorial Pacific SST over the last millennium (Emile-Geay et al., 2013b). More recently, the PAGES (PAst Global ChangES) Ocean2k project has used available coral $\delta^{18}O$ and Sr/Ca records (39 sites) to reconstruct regional tropical SST for the past four centuries at annual resolution (Tierney et al., 2015a), and further efforts are underway to address the reconstruction of the spatiotemporal patterns of $\delta^{18}O_{sw}$ (McGregor et al., 2016).





With regard to hydroclimate variability and change during the CE, the few available $\delta^{18}O_{sw}$ reconstructions derived
from paired coral Sr/Ca and $\delta^{18}O$ that extend back for a century or more suggest freshening in the southwestern tropical Pacific
around the mid-19th century (Hendy et al., 2002; Ren et al., 2003; Linsley et al., 2004), in the northwestern subtropical Pacific
after 1910 CE (Felis et al., 2009a), and in the central tropical Pacific after 1970 CE (Nurhati et al., 2011b)(Figure 1). No $\delta^{18}O_{sw}$
trend is uncovered in the southwestern tropical Indian Ocean during the 20th century (Zinke et al., 2008), whereas saltier surface
waters are inferred in the northwestern tropical Atlantic after 1950 CE (Hetzinger et al., 2010). In all cases, the reconstructions
reveal pronounced interannual and decadal-to-multidecadal variability in $\delta^{18}O_{sw}$. While some reconstructed changes in $\delta^{18}O_{sw}$
can be definitively linked to changes in the surface freshwater balance related to ENSO extremes (e.g. Nurhati et al., 2011b), the
dynamical drivers of inferred lower-frequency changes in $\delta^{18}O_{sw}$ remain unclear in most cases. Efforts to quantify and describe
variations in $\delta^{18}O_{sw}$ have thus far relied on scant observational data available from limited monitoring campaigns (Fairbanks et
al., 1997; Benway and Mix, 2004; McConnell et al., 2009; Conroy et al., 2014) compiled into a global database (Schmidt et al.,
1999; LeGrande and Schmidt, 2006).

Despite existing uncertainties and challenges, $\delta^{18}O_{sw}$ is closely related to ocean salinity, which itself has been referred
to as "nature's rain gauge" for the data-sparse expanses of the ocean (Terray et al., 2011). Unlocking the full potential of coral
$\delta^{18}O_{sw}$ reconstructions would perhaps enable researchers to resolve natural versus anthropogenic trends in the hydrological
cycle. The marine hydrological cycle and the terrestrial hydrological cycle are also importantly linked through atmospheric
moisture transport and continental runoff (e.g. Gordon, 2016). Moisture evaporating from the ocean feeds terrestrial
precipitation, and recent studies indicate that interannual changes in the ocean-to-land moisture transport leave an imprint on sea
surface salinity, establishing the potential to improve seasonal rainfall predictions in regions such as the Sahel and the
Midwestern United States (Li et al., 2016a, b). Coral $\delta^{18}O_{sw}$ reconstructions also can provide marine perspectives on past
ocean-atmosphere-land interactions and on terrestrial hydroclimate variability and change observed in continental-scale tree-
ring networks (e.g. Cook et al., 2015b). The ultimate potential of these applications is further enhanced by a new generation of
ocean, atmosphere, and fully-coupled models equipped with water isotope tracer modules (see Sections 3 and 4), which promise
to deliver additional insights on the dynamical controls – both oceanic and atmospheric – on $\delta^{18}O_{sw}$ variations.

*2.2 Tree-rings*

Tree-ring proxies are an important source of high-resolution, absolutely dated information about the hydroclimate of the CE.
They are widespread, well-replicated, and can be statistically calibrated against the overlapping instrumental records to produce
validated reconstructions and associated estimates of uncertainty of past climate variability at annual resolution. Annual ring
width is the most common proxy measurement, but sub-annual increments (Griffin et al., 2013; Knapp et al., 2016), wood
density (Briffa et al., 1992b), the Blue Intensity proxy of wood density obtained from high resolution images (McCarroll et al.,
2002; Campbell et al., 2007; Wilson et al., 2014), and stable and radiogenic isotopes (Suess, 1980; McCarroll and Loader, 2004;
Gagen et al., 2011a) also provide information from tree-ring archives about past environmental conditions. The process of
developing tree-ring proxy time series begins with collecting multiple increment cores from multiple trees at a site. The annual
rings are exactly dated, using a process of visual and statistical cross-comparison and correlation called crossdating (Douglass,
1941; Stokes and Smiley, 1968; Fritts, 1976; Holmes, 1983). All dating issues are resolved prior to measurement, which is at
high resolution (0.01 or 0.001 mm) and with high precision. Statistical validation of the dating is performed using the software
COFECHA (Holmes, 1983). Tree-ring width, sub-annual increment and density series typically consist of dozens of trees with
multiple cores per tree. More labor-intensive or costly methods using isotopes may be derived from fewer series. Proxy
measurements on individual cores are combined into mean time series called 'chronologies.' The similarity among individual




series is assessed using the interseries correlation (the mean correlation between individual series and the other series) and the

Expressed Population Signal (EPS), which estimates how well a sample reflects the signal of a hypothetical population based on

the number of samples and the interseries correlation (Wigley et al., 1984; Cook and Kairiukstis, 1990).

Tree-ring proxy measurements reflect a combination of internal and external influences on tree growth that include

climate, but also tree age and geometry and disturbance. Cook (1987) proposed a Linear Aggregate Model of Tree Growth, a

heuristic model to describe the different influences on the growth of annual rings:

$$R_t = A_t + C_t + D_t^i + D_t^e + \varepsilon_t$$

where $R_t$ is the width of the ring in year $t$, $A_t$ is growth imposed by tree age and geometry, $C_t$ is the influence of climate, $D_t$

represents the impact of tree- ($^i$) and site-level disturbance ($^e$), respectively, and $\varepsilon_t$ is variability not captured by the other terms,

including random error. The influence of climate is not necessarily limited to the year or growing season associated with the

ring, as prior-year climate and growth can influence subsequent ring formation (Fritts, 1966). For application of tree-ring

proxies to paleoclimatology, we seek to isolate the climatic influence on tree-ring growth, remove age-related growth trends,

and minimize the influence of disturbance, such that past climate as a function of ring width can be estimated.

The global network of tree-ring chronologies records a diverse set of seasonal climate signals (Meko et al., 1993;

Williams et al., 2010; Breitenmoser et al., 2014; St. George, 2014; St. George and Ault, 2014; Wise and Dannenberg, 2014).

The network itself is biased toward terrestrial mid- and high-latitudes in the Northern Hemisphere (NH), reflecting a mix of

historical focus, continental forest regions, and the abundance of trees that form unambiguous annual growth rings. Tropical

regions in particular (Stahle, 1999), as well as the Southern Hemisphere (SH) remain under-represented, as does sub-Saharan

Africa (Jones et al., 2009). High latitude and high elevation trees near the limit of their thermal tolerance are more likely to

record variations in growing season temperature, while trees located in dry and semi-arid sites reflect predominantly moisture

availability. Wood density proxies typically have high correlations with summer temperatures at high latitude sites (Briffa et al.,

2002a, b), although high-latitude locations away from treeline may also reflect precipitation and moisture (Hughes et al., 1994).

Tree-ring growth within temperate mesic forests of the mid-latitudes often reflects species-specific or mixed environmental

signals including both moisture and temperature (Cook and Jacoby, 1977; Graumlich, 1993; Meko et al., 1993; Pederson et al.,

2004; Babst et al., 2013), with some notable exceptions (e.g. Stahle et al., 1998).

The majority of tree-ring proxies reflect seasonal rather than annual climate variability. For moisture-sensitive trees,

the climate response can vary across regional and continental scales. In Mediterranean regions and the American Southwest,

winter, spring, and water-year precipitation usually dominate moisture variability and tree-ring formation, whereas continental

and mesic forests reflect summer or growing season precipitation (Meko et al., 1993; Touchan et al., 2005; Esper et al., 2007;

Buentgen et al., 2010; St. George et al., 2010; Buentgen et al., 2011; St. George and Ault, 2014; Touchan et al., 2014). Tree-

ring proxies may also record winter snow accumulation, either via the subsequent influence of spring melt on growing season

water balance or more directly via limitations on the initiation of growth (Pederson et al., 2011; Coulthard and Smith, 2016).

Tree-rings also have been used with considerable success to reconstruct riverflow (e.g. Stockton and Jacoby, 1976; Woodhouse

et al., 2006; Meko et al., 2007; Maxwell et al., 2011; Harley et al., 2017) and runoff ratio (Lehner et al., 2017), as tree growth

can reflect an integrated seasonal moisture balance signal in a similar fashion as watersheds. Sub-annual chronologies, isolating

earlywood and latewood width, can provide differential seasonal resolution. In the North American monsoon region, for

instance, earlywood width reflects winter-spring precipitation, while latewood width can more strongly reflect the summer rains

associated with the monsoon (Meko and Baisan, 2001; Stahle et al., 2009; Griffin et al., 2013). Stable carbon and isotope

chronologies are less common than ring width or wood density series, but can reflect a host of hydroclimatic processes,

including drought (Treydte et al., 2007; Andreu-Hayles et al., 2016), cloud cover (Young et al., 2010; Gagen et al., 2011b),



relative humidity (Edwards and Fritz, 1986), rainfall amount (Evans and Schrag, 2004), snow (Treydte et al., 2006), vapor pressure (Voelker et al., 2014), and moisture source and atmospheric circulation (Saurer et al., 2012; Williams et al., 2012).

Ring-width proxies of hydroclimate in many cases reflect soil moisture as opposed to rainfall directly. Metrics that model the integrated effect of precipitation, evapotranspiration, and soil water processes on moisture balance may account for more variability in ring width than precipitation alone (e.g. Kempes et al., 2008). More importantly, use of these metrics as climate reconstruction targets, including the Palmer Drought Severity Index (PDSI) and Standardized Precipitation Evapotranspiration Index (SPEI) also permit large-scale uniform field reconstructions of droughts and pluvials even in the

presence of a seasonality in rainfall sensitivity or mixed climate response (Cook et al., 1999). Franke et al. (2013) and Bunde et al. (2013) raised concerns about the spectral fidelity of tree-ring proxies, observing that the slope of their frequency continuum resembled neither that of temperature nor precipitation; however, a re-examination of the coeval spectra of a global network of tree-ring chronologies presented here in Figure 2 suggests that this apparent discrepancy is because hydroclimate tree-ring proxies reflect soil moisture, which displays greater power at decadal to centennial frequencies.

Isolating the climate signal itself from the tree-ring proxy measurements also requires removing or accounting for age-related or geometric growth trends and minimizing the influence of disturbance or other ecological processes – a method known as detrending and standardization. As Cook (1987) noted, the age-related component of tree-ring width series is "a non-stationary, stochastic process which may, as a special case, be modeled as a deterministic process." In practice, this means that although geometric curves can be fit to tree-ring series and used to remove these trends, in most cases the true shape of the

curve remains uncertain. Of concern for climate reconstruction is the removal of low-frequency climate information or the potential for systematic non-climatic biases to persist or be imparted due to detrending and standardization. The choice of growth curve for removing the effects of tree age and geometry may have considerable influence on the low-frequency component of the chronology. Detrending by means of using individual curves fit to measured tree-ring width series unavoidably results in the 'segment length curse' (Cook et al., 1995), which mathematically constrains the resolved frequency

range of the resulting chronology to be no lower than that corresponding to the mean length of the individual series. In order to exorcise this curse, dendroclimatologists have increasingly used regional curve standardization (RCS), which detrends each series using an estimated common growth curve for all individual trees (Briffa et al., 1992a; Esper et al., 2002) and can preserve variance at periods longer than the detrended measurement series. Nevertheless, the RCS method may impart its own biases (Briffa and Melvin, 2011; Anchukaitis et al., 2013). Similar concerns may exist for newer 'signal free' detrending methods that

attempt to avoid bias in trends due to medium-frequency variance (Melvin and Briffa, 2008). Whichever method is used, detrending and the removal of non-climatic influences continues to represent one of the most important 'known unknowns' (Logan, 2009) in the reconstruction of past hydroclimate from tree-ring proxies.

### 2.3 Speleothems

Speleothems (e.g. stalagmites, stalactites) are cave deposits that form when calcium carbonate precipitates from degassing solutions as they seep into limestone caves, leaving behind archives that can provide absolutely dated records of terrestrial hydroclimate over the CE from terrain that contains karst deposits. Dates of the carbonate material employ disequilibrium U/Th dating that targets intermediate daughters of the 238U decay chain, specifically 230Th and 234U, to provide absolute chronologies for stalagmite-based paleoclimate records (Edwards et al., 1987; Cheng et al., 2000) with precisions of <1% (2

sigma; Shen et al., 2012) over the last ~650 thousand years (e.g. Cheng et al., 2016b). Stalagmite-based paleoclimate records typically resolve millennial to orbital-scale variations of climate at a given site, owing to their slow and often steady growth averaging 2-20 microns/yr over many tens of millennia. Currently, numerous absolutely-dated, well-replicated, decadally- to



centennially-resolved stalagmite $\delta^{18}$O records exist from South America (Cruz et al., 2005; Wang et al., 2007), the Western Pacific (Partin et al., 2007; Griffiths et al., 2009; Griffiths et al., 2010; Meckler et al., 2012; Carolin et al., 2013), China (Wang et al., 2001; Zhang et al., 2008; Cheng et al., 2016a), and the Middle East (Bar-Matthews et al., 1997; Bar-Matthews et al., 1999; Cheng et al., 2015).

The generation of sub-annual to annually-resolved stalagmite records holds immense potential to reconstruct hydroclimate conditions in many regions through the CE. In most cases, such high-resolution records rely on 10-500 micron sampling of unusually fast-growing stalagmites that form on the order of 100-2000 microns/yr (Treble et al., 2003; Partin et al., 2013; Chen et al., 2016). Even for these fast-growing records, however, the period of the CE may only span several centimeters to, in exceptional cases, many tens of centimeters, ultimately limiting the number of 1-2 mm-scale samples that can be drilled for conventional, high-precision U/Th dating. In many cases, stalagmite growth rates also vary significantly over the CE, and/or growth can slow to near-zero values, causing a hiatus that can be poorly resolved by the relatively small number of available U/Th dates. These circumstances represent a significant challenge to the generation of age models for stalagmites spanning the CE based on radiometric dates. Two programs – BChron (Haslett and Parnell, 2008) and StalAge (Scholz and Hoffmann, 2011) – allow researchers to calculate age models and their uncertainties for stalagmite records given a set of radiometric dating constraints, including the identification of potential hiatuses (see Scholz et al. (2012) for a review of age modeling approaches to speleothem records). In rare cases, annually-banded stalagmites afford the generation of layer-counted chronologies that can be tested against radiometric ages (Polyak et al., 2001), although recent work has demonstrated that apparent annual banding in stalagmites may not always be strictly annual (Shen et al., 2013).

The most widely used and best understood measurement in speleothem climate reconstructions is the oxygen isotopic composition, or $\delta^{18}$O, of calcite (e.g Fleitmann et al., 2004), which under constant precipitation conditions reflects changes in cave temperature as well as changes in the $\delta^{18}$O of the cave dripwater feeding the stalagmite. Over the CE, air temperature in a given cave likely changed very little (<1°C or ~0.2‰ in stalagmite $\delta^{18}$O units) such that the observed speleothem $\delta^{18}$O variations of up to 1‰ are governed primarily by groundwater $\delta^{18}$O variability. In most scenarios, groundwater $\delta^{18}$O composition reflects a weighted mean of rainfall $\delta^{18}$O averaged over the preceding months in wet environments (Moerman et al., 2014), while it may reflect years in semi-arid and arid environments (Ayalon et al., 1998). Some studies also show that recharge of the aquifer occurs only during months when rainfall passes a given threshold, typically during the wet season in the tropics (e.g. Jones and Banner, 2003; Partin et al., 2012), but possibly associated with the winter storm season in the extratropics. While other indicators such as band thickness (e.g. Rasbury and Aharon, 2006; Asmerom et al., 2007), trace metal ratios (see Fairchild and Treble, 2009 and references therein), and carbon isotopes (Fairchild et al., 2000; Frappier et al., 2002; Oster et al., 2015) reflect hydroclimate variability at certain sites and continue to be developed, speleothem $\delta^{18}$O remains the primary hydroclimate proxy.

Modern studies of rainfall $\delta^{18}$O, cave dripwater $\delta^{18}$O, rainfall amount, and speleothem $\delta^{18}$O show that, depending on the location, speleothem $\delta^{18}$O may be a record of local rainfall amount, regional hydroclimate variability, and/or changes in the source of the moisture (which is linked to regional hydroclimate variability) (e.g. Flohr et al., 2017 and references therein). In the past, the so-called "amount effect" was invoked to interpret a speleothem $\delta^{18}$O record (Dansgaard, 1964), which assumed an inverse, empirical relationship between rainfall amount and rainfall $\delta^{18}$O (Rozanski et al., 1993; Gat, 1996). Over the tropical ocean, or over small tropical islands, the "amount effect" dominates the relationship on monthly to interannual timescales (Kurita, 2013), whereby local rainfall $\delta^{18}$O reflects the cumulative fractionation of water isotopes over the transit of the vapor parcel through space and time. As such, many calibration efforts demonstrate that speleothem $\delta^{18}$O is inversely correlated with



the instrumental rainfall at the site on annual to decadal timescales (e.g. Partin et al., 2013). In other locations, rainfall $\delta^{18}O$ and cave dripwater $\delta^{18}O$ correlate better with indices of large-scale circulation patterns (e.g. the El Niño-Southern Oscillation (ENSO) index) than with local rainfall amount (Moerman et al., 2013; Moerman et al., 2014). Lastly, some studies show that

speleothem $\delta^{18}O$ can reflect changes in the source of precipitation, which alters the $\delta^{18}O$ of rainfall (Aggarwal et al., 2004; Breitenbach et al., 2010). Unified frameworks and theories are now being tested to determine the underlying mechanisms controlling rainfall $\delta^{18}O$ in order to more directly compare rainfall output from climate models with speleothem $\delta^{18}O$ records (e.g. Lewis et al., 2010; Aggarwal et al., 2012; Jones et al., 2016; Tharammal et al., 2017). These efforts are also relevant for other proxy records of precipitation $\delta^{18}O$ and $\delta D$, for example sediments (Section 2.4) and coral records that are sensitive to the

isotopic composition of rainfall (Section 2.1).

In addition to whether speleothem $\delta^{18}O$ records rainfall amount, source, or other large-scale processes, there are other challenges that arise when interpreting them. A recent paper by Hu et al. (2017) points out several challenges that especially apply over the CE, which include: serial autocorrelation, the test-multiplicity problem in connection with a climate field, and the presence of age uncertainties; Hu et al. (2017) include information and code on how to address these issues when interpreting a

record. Additionally, proxy system models (PSMs; see Section 4 for more discussion) are being used to simulate depositional processes, that can incorporate karst processes that may redden the karst signal (Dee et al., 2015). Such studies are indicating that even in the simplest of karst models, signal reddening and the impacts of high-to-low frequency signal enhancement can be captured and explained (Partin et al., 2013).

Finally, the expanding network of speleothem records is being collected and characterized by the PAGES2k Trans-

Regional Project, Iso2k (Partin et al., 2015). This effort has identified over 65 published records of speleothem-based hydroclimate estimates over the CE. These records span every major continent outside Antarctica. Continuing coordinated efforts to generate well-replicated, high-resolution records of speleothem $\delta^{18}O$ that span all or part of the CE will yield robust reconstructions of hydroclimate that can be directly compared to the expanding number of simulations spanning the CE, some of which are isotope-equipped (and more are expected as part of the CMIP6 archive).


*2.4 Sediments*

Marine and lacustrine sediment cores play an important role in the reconstruction of past hydroclimate variability over the CE, as they often extend further back in time than corals and tree rings. There also are typically multiple hydroclimate proxies that can be measured in any given sediment core, theoretically allowing for better isolation of the climatic signal from noise. Such

proxies include:

*Lake level indicators*: Inference of past lake levels, particularly from closed-basin lakes, serves as a sensitive indictor of P-E and therefore regional moisture balance (c.f. Verschuren et al., 2000; Shuman et al., 2009; Goldsmith et al., 2017).

*Pollen and other microfossil transfer functions:* Microfossil assemblages such as pollen and diatoms can be directly regressed against modern climatic variables such as P-E or mean annual precipitation. A number of statistical approaches may

be employed, such as the modern analog technique (e.g. Overpeck et al., 1985) artificial neural networks (e.g. Peyron et al., 1998), and hierarchical Bayesian models (Haslett et al., 2006).

*Runoff indicators*: Physical and chemical characteristics of lake and marine sediments such as the concentration of major elements (Ti, Fe, Ca) or grain size can be used to infer runoff intensity, which is in turn related to the intensity, frequency, or amount of rainfall. Scanning XRF techniques make it possible to analyze such characteristics at very high temporal

resolution (e.g. Haug et al., 2001; Haug et al., 2003; Moreno et al., 2008).



*Proxies for water isotopes*: In marine cores, measurement of the $\delta^{18}O$ of surface-dwelling foraminifera provides insight on hydroclimatic change over the ocean, just as $\delta^{18}O$ does in corals (see Section 2.1). Simultaneous measurement of the Mg/Ca ratio in foraminifera gives an independent constraint on temperature, allowing for isolation of the signal associated with $\delta^{18}O$ of seawater (Elderfield and Ganssen, 2000). In both marine and lacustrine sediments, measurements of the hydrogen isotopic composition of lipids from aquatic algae and terrestrial higher plants can be used to reconstruct (sea or lake) water $\delta D$ and precipitation $\delta D$, respectively (e.g. Sachs et al., 2009; Tierney et al., 2010; Richey and Sachs, 2016). Such proxies provide important constraints on processes that influence isotopes of precipitation, such as rainfall amount, source, and seasonality.

The diverse and independent proxy systems available are a clear strength of sedimentary archives. Nevertheless, working with sediment records on recent timescales presents a number of unique challenges that must be considered when interpreting data and incorporating them into a multi-proxy reconstruction framework. An obstacle inherent to working with sediments is the uncertainty related to chronology – that is, the assignment of depth horizons to points in time. Varved sediments (sediments in which distinct layers are deposited on a demonstrably annual basis) have relatively tightly-constrained chronologies, although counting errors will propagate downcore, as is the case with any annually-layered archive that is not cross dated (Comboul et al., 2014). Varves are nevertheless rare in marine environments as well as in the terrestrial tropics, and in their absence chronological assignment depends on radiometric methods. The two most common systems used in dating recent marine sediments are $^{210}Pb$ and $^{14}C$. With a half-life of 22.2 years, $^{210}Pb$ is ideal for dating sediments spanning the last 100-150 years and can provide an accuracy ranging from 1-20 years. Older sediments are dated primarily with $^{14}C$, which has a half-life of 5,730 years. In lakes, $^{14}C$ may be measured on terrestrial macrofossils or bulk organic carbon, although the latter can be affected by the presence of old carbon (coming from, for example, an isolated hypolimnion or terrestrial $CaCO_3$ input). In marine sediments, $^{14}C$ is typically measured on species of planktonic foraminifera. In this case, one must account for the $^{14}C$ age of the ocean water (ca. 200-1500 years), or the marine "reservoir effect." Globally, the average $^{14}C$ age of ocean water is 400 years (Stuiver and Braziunas, 1993). Local deviations from this age are commonly expressed as $\Delta R$, the values of which are determined by measuring the $^{14}C$ age of contemporary (known age) pre-nuclear marine specimens like bivalves and corals.

Uncertainties in $^{14}C$ dating range from decades (ca. 30-40 years) to centuries (ca. 200-300 years) and depend on the accuracy of the measurement, where the date falls on the $^{14}C$ calibration curve (placement on a radiocarbon production "plateau" will increase error) and uncertainties in reservoir corrections (see example in Figure 3a). Thus, over the timeframe of the CE, chronological uncertainties can be formidable. Increased density of dating and independent constraints on sedimentation rates can improve the precision of the chronology, but multiple age-depth models are always possible. This uncertainty is best dealt with through an ensemble approach; i.e. use of a Monte Carlo or Bayesian age modeling method to produce a posterior ensemble of age-depth models that can then be used iteratively in a reconstruction framework (Ramsey, 2008; Blaauw and Christen, 2011; Tierney et al., 2013; Werner and Tingley, 2015).

Another special consideration for sedimentary archives is the role of bioturbation. In low-oxygen environments, sediments may be deposited essentially "undisturbed" and will appear laminated. In most locations, however, bottom-water oxygen is present and the benthos (organisms living on and in the seafloor) will mix the upper layers of the sediment. In the ocean, bioturbation depths average 8 cm globally, but vary from 0-20 cm depending on productivity, bottom water oxygen, and sedimentation rate (Teal et al., 2008). Bioturbation thus acts as a low-pass filter on sedimentary proxy signatures and will redden the spectra of proxy time series. Diffusion-based forward models for bioturbation exist and can be incorporated into reconstruction frameworks (e.g. Trauth, 2013).

The proxy systems in sediment archives also have the capability to redden the spectra of paleoclimate signatures, and this must be considered both for inter-archive and proxy-model data comparisons. Lake level records, for example, are typically



low-passed representations of regional hydroclimate. Lake water residence time and hydraulic considerations buffer variability in P-E, such that lake levels will have more power at low frequencies, and less at the high-frequencies, than the local climate variable (see example in Figure 3b). The amount of reddening depends on individual lake systems and can be forward modeled if lake geometry, inputs, and outputs are known (e.g. Hurst, 1951; Hostetler and Benson, 1994).

Complications arising from chronology and bioturbation mean that most sediment proxy data are best suited for inferring hydroclimate change on multidecadal and longer timescales. Nevertheless, understanding shifts in hydroclimate over these longer intervals during the CE is important, as such changes set the baseline for long-term droughts or pluvials, and complement higher-resolution perspectives from other hydroclimatic proxies.

*2.5 Documentary evidence*

Documentary evidence includes non-instrumental information on past climate and weather conditions before the advent of continuous meteorological measurements, the principal sources of which are descriptive documentary data (i.e. descriptive observations of weather, reports from chronicles, ship logbooks, travel diaries etc.) and documentary proxy data. This latter source of information refers to more indirect records of events or practices tied to specific weather events or climatic conditions;

examples include the beginning of agricultural activities, the freeze and thaw dates of waterways, records of floods, and reports of religious ceremonies in response to impactful meteorological conditions (see Brazdil et al., 2005 for a review).

    Descriptive evidence generally has good dating control and high temporal resolution (for specific periods and regions even daily resolution is possible), but this evidence is also typically discontinuous. Documentary sources also often emphasize extreme events because the consequences of their socioeconomic impacts render them more likely to have been recorded

(Brazdil et al., 2005). Typical examples include hydro-meteorological extremes (flooding, hail, torrential rains) and other natural hazards that impacted the success of harvests or placed livestock in jeopardy (Pfister et al., 1999; Camenisch et al., 2016). China, Japan and Europe comprise the three main geographic regions where an abundance of historical-climatological sources exist. More recently, assessments have been initiated for records from South America that date back to the beginning of Spanish colonization (Neukom et al., 2010). Data from Africa, North America and Australia are available for much shorter

periods into the past, in comparison with other regions where more abundant documentary data exist (e.g. Grab and Nash, 2010; Nash and Grab, 2010; Fenby and Gergis, 2013; Gergis and Ashcroft, 2013; Neukom et al., 2014; Nash et al., 2016).

    *2.5.1 Europe*

Research on documentary-based past hydroclimate in Europe is temporally and geographically unbalanced (e.g. Brazdil et al.,

2005; Pauling et al., 2006). Potentially useful documentary evidence is available in most of Europe and the Mediterranean regions, although only a few regions include hydroclimatic studies (including droughts and floods and other extremes), these being in the Czech Republic, Germany, the eastern Mediterranean, Byzantium, Hungary, Italy, Portugal, Spain, Switzerland and in the U.K. (Brazdil et al., 2005; Luterbacher et al., 2006; Brazdil et al., 2010; Wetter et al., 2011; Brazdil et al., 2012; Brazdil et al., 2013; Wetter et al., 2014; Brazdil et al., 2016; Xoplaki et al., 2016). New promising data from other regions do exist, but

they have not yet been fully explored. Some records prior to 1000 CE (e.g. Xoplaki et al., 2016 and references therein) are available during the Byzantine Empire (including the Balkans) and the Carolingian Empire. Subsequent to 1000 CE, specific periods can be characterized (Brazdil et al., 2005; Pfister et al., 2008) that include a progression from individual reports of significant socioeconomic anomalies and disasters (weather induced, floods, droughts) from 1000-1200 CE to almost full descriptions for monthly weather (including some daily weather and extremes) from 1500-1800 CE. Starting in about 1650

through 1860 CE, early instrumental measurements made by individuals and organized by scientific and economic societies



become available. These earlier efforts are followed by short-lived international instrumental networks (up to the end of the 18th century) and initiatives within 19th-century emerging nation states.

Despite the relative abundance of information in Europe, our understanding of hydroclimate variability (including short-duration and extensive drought and flood periods) is still limited because only a small number of well-dated documentary
proxy records with high temporal resolution are available and unevenly distributed over the continent (e.g. Brazdil et al., 2005). Some synthesis efforts nevertheless have been pursued. Pauling et al. (2006) combined early instrumental series, documentary proxy time series and natural proxies to reconstruct seasonal precipitation fields for European land areas covering the past half millennium. A similar reconstruction was provided by Casty et al. (2007) that used only documentary and natural proxies tied to precipitation, which allows comparison to independently reconstructed temperature and pressure fields. Collectively,
identifying and characterizing historical extreme events that have severely stressed human or natural systems (e.g. the onset, duration, frequency, and intensity of droughts and floods) is a critical endeavor, and our capacity for studying these events is possible through the continued development and interpretation of documentary records in Europe.

*2.5.2 Asia*

Hydroclimatic reconstructions over China, including those from documentary records, are reviewed in Ge et al. (2016), Fan (2015) and Hao et al. (2016). The region has rich historical resources including classical documents, local gazettes, governmental archives, and personal diaries (Figure 4). This collection of information has been extensively studied to extract weather/climate information in efforts that extend back to the 1920s (e.g. Chu, 1926; Yao, 1943; Chu, 1973; Zhang and Gong, 1980). Abundant hydroclimate reconstructions from these records can be separated into two common classes: 1) qualitative
reconstructions such as dryness and wetness (D-W) indices, and 2) quantitative precipitation reconstructions. Qualitative reconstructions are generated by counting the number of dry/wet events, which occasionally consider the timing, scale and severity of these events based on written descriptions in a wide range of historical papers (Gong and Hameed, 1991; Ge et al., 2016). Numerous reconstructions in this class have been published, for instance D-W series for 120 sub-regions across China from 1470-2000 CE (CMA, 1981; Zhang et al., 2003), a regional D-W dataset from 960-1992 CE (Zhang et al., 1997), the
history of moisture conditions for the last 2000 years (Gong and Hameed, 1991), D-W series for 63 sites since 137 BCE (Zhang, 1996), and a relatively new D-W dataset for the period 501–2000 CE that has been subsequently extended to cover the last 2000 years (Zheng et al., 2006; Zheng et al., 2014). Characteristics of drought and floods based on these records have been extensively studied, such as the frequency, severity, spatial patterns, and decadal-to-centennial variability of hydroclimate, as well as the relationship between D-W indices and temperature (e.g. Wang and Zhao, 1979; Zhang and Crowley, 1989; Yan et
al., 1992; Qian et al., 2003; Shen et al., 2009; PAGES 2k Consortium, 2013).

For quantitative reconstructions, around 300 years of annual/seasonal precipitation time series are available in north China and the mid-lower Yangtze River valley (Zhang and Liu, 2002; Zhang et al., 2005; Zheng et al., 2005) using accurate weather/climate descriptions from "Qing Yu Lu" (Clear and Rain Records), "Yu Xue Fen Cun" (rainfall infiltration and snowfall depth, Ge et al. (2005)), and others. Qualitative precipitation reconstructions have been used to study Meiyu (plum
rain) activity (Zhang and Wang, 1991; Ge et al., 2008a; Ding et al., 2014a), an exceptional flooding event in 1755 CE (Zhang et al., 2013), and East Asian Summer Monsoon variations at multi-decadal and centennial scales (Hao et al., 2015). Because quantitative precipitation reconstructions over China are relatively short and spatially limited, researchers have tried to provide quantitative precipitation information by combining D-W indices and other high-resolution proxies. For example, incorporating D-W indices and tree-ring proxies, Yi et al. (2012) reconstructed annual summer precipitation in north-central China back to
1470 CE, and Feng et al. (2013) generated a gridded reconstruction of warm season precipitation over Asia spanning the last



500 years. Some of these Chinese efforts are complemented by efforts in Japan – many documents dating from the Edo era contain records of daily weather conditions. Most of the known daily weather records have been digitized and added to the Historical Weather Database of Japan (Mikami, 1988, 2008).

**3. Coupled Model Simulations of the Common Era**

In concert with hydroclimate proxy development over the last several decades, efforts to model the CE have also progressed considerably. In the last decade, two complementary approaches have enhanced the relevance of paleoclimate simulations for our understanding of past climate evolution. The first is the use of single-model ensembles (Jungclaus et al., 2010; Hofer et al., 2011; Otto-Bliesner et al., 2016) comprising multiple simulations with the same model configuration and experimental design (i.e. Schmidt et al., 2011; Otto-Bliesner et al., 2016). These ensemble frameworks allow for systematic characterization of simulated internal climate variability and forced responses, as well as an assessment of uncertainties in boundary forcing estimates. Single-model ensembles are nevertheless subject to the peculiarities and deficiencies of the individual model used.

The second development has been coordinated multi-model ensembles, such as the PMIP last-millennium simulations (Braconnot et al., 2012; Schmidt et al., 2014; Jungclaus et al., 2016), that allow the combination of model information either probabilistically or via measures of multi-model consensus. The incorporation of the core PMIP simulations into the framework of CMIP has allowed for cross time-period analyses of past, present, and future climate states and variability. Moreover, the fidelity of CMIP models is thoroughly evaluated over the historical instrumental period (e.g. Flato et al., 2013), and because the same models are used for simulations of the past, their strengths and limitations can be taken into account when comparing with paleoclimatic information. Collectively, the analyses of single-model ensembles, multi-model ensembles and individual models can provide insight into how models simulate climate dynamics on paleoclimatic timescales, informing our understanding of the real climate system and improving our assessments of climate risks in the future. For a list of available simulations of the last millennium, including the associated forcings, see Table 1.

*3.1. Simulating Hydroclimate over the Common Era*

In general, climate models do not simulate hydroclimatic variables, such as precipitation, as realistically as surface temperatures (Flato et al., 2013; e.g. Figure 5a). This is due to limitations in the representation of hydroclimatic impacts associated with modes of variability (Davini and Cagnazzo, 2014; see Figure 5b) and processes that must be parameterized such as convection and cloud physics (Stephens et al., 2010). Nevertheless, models are generally successful at simulating the large-scale atmospheric circulations that drive areas of subsidence and uplift, and thus generally represent well the zonal mean hydroclimate, particularly in the multi-model ensemble mean (Flato et al., 2013). This is less true of regional hydroclimate dynamics, as many models struggle to reproduce the full characteristics of observed phenomena such as the global monsoon systems (e.g. Geil et al., 2013 for the North American Monsoon; Sperber et al., 2013 for the Asian Monsoon), some midlatitude storm tracks and the frequency of associated cyclonic systems (e.g. in the western and north Atlantic - Colle et al., 2013; Dunn-Sigouin and Son, 2013; Sheffield et al., 2013; Zappa et al., 2013). A full consideration of hydroclimate also includes representation of land surface processes. While hydrology, vegetation and soil moisture modeling has become increasingly complex, these components of models have not been as widely validated and paleoclimate simulations often do not include important processes such as dynamic vegetation (Braconnot et al., 2007; Braconnot et al., 2012).

Because of the multiple model challenges listed above, paleoclimate information plays a critical role in evaluating model fidelity, as it provides the only source of 'out-of-sample' validation information (see Section 4). Model assessment is performed by directly comparing reconstructed and simulated time series or fields, by studying statistics of variability and



extreme events, and by analyzing important processes, such as the simulated responses to changes in external forcing. The limited spatial and temporal resolution of the paleoclimate record (i.e. relative to the observational record), nevertheless leads to important challenges associated with explicitly characterizing model deficiencies in simulating climate dynamics on paleoclimate timescales. Despite these challenges, existing studies have demonstrated important uncertainties associated with
hydroclimate comparisons between paleoclimate information and models that illustrate the importance of future work on the subject.

At the continental scale, Ljungqvist et al. (2016) presented a spatial reconstruction of NH hydroclimate variability for the last 1200 years and compared the data with PMIP3 last-millennium simulations. They discuss the relationship between the long-term temperature evolution and hydroclimate and diagnose a statistically significant co-variability for particular regions,
but also widespread deviations from this relationship. In particular, the authors find that reconstructions and simulations agree reasonably well over the pre-industrial interval and indicate that the 13[th] century had the most extensive dry conditions. Attribution to external forcing could, however, not be obtained from the assessment.

Models are also capable of reproducing multidecadal drought periods in some arid regions, a prominent and socio-economically relevant feature of past hydroclimate variations. For instance, Coats et al. (2015b) analyzed megadroughts in the
American Southwest in proxy reconstructions and in PMIP3 simulations, finding that models can reproduce events that are similar in extent and severity to tree-ring derived reconstructions. The authors nevertheless also describe pronounced differences between paleoclimatic reconstructions and models. For example, not all of the PMIP3 models associate megadroughts with teleconnections originating in the tropical Pacific Ocean, which is the assumed origin in the real climate system, and therefore may not accurately represent the dynamics underlying real-world megadroughts (Coats et al., 2016a;
Coats et al., 2016b). Whereas Coats et al. (2015b) diagnose the general ability of climate models to reproduce low-frequency hydroclimate variability from a dynamical perspective, other studies based on statistical rescaling (Ault et al., 2012) find that model simulations underestimate low-frequency (in the 50- to 200-year range) precipitation variability. Spectra of reconstructed hydroclimatic variables from Western North America show a considerably "redder" characteristic than those from models (Ault et al., 2013a). These findings are also tied to interpretations of proxy spectral fidelity, as discussed in Section 2.


*3.2. Volcanic Forcing of Hydroclimate*

Volcanic eruptions are the most important external radiative forcing at the global scale during the pre-industrial interval of the CE. Although the impacts persist only for a few years, volcanic forcing from multiple events constitutes the most important contributor to the global centennial-scale cooling observed from 850-1800 C.E., based on comparisons of single
forcing experiments (Otto-Bliesner et al., 2016) or through the use of all-forcing simulations in which the contributions from individual forcings are isolated offline (Atwood et al., 2016). For this reason, volcanic eruptions are particularly important for gauging the forced hydroclimatic response in last-millennium simulations.

Both observational and modeling studies consistently find a decrease in global precipitation following large explosive eruptions; the main regions experiencing decreased precipitation tend to be the tropics (Robock and Liu, 1994; Yoshimori et al.,
2005; Trenberth and Dai, 2007; Schneider et al., 2009) and monsoon regions (Schneider et al., 2009; Joseph and Zeng, 2011; Wegmann et al., 2014). Iles and Hegerl (2014) further examined the precipitation response to volcanic eruptions in the CMIP5 historical simulations compared to three observational datasets. Global precipitation significantly decreases following eruptions in CMIP5 models, with the largest decrease in wet tropical regions. Iles et al. (2013) examined the global precipitation response to large low-latitude volcanic eruptions using an ensemble of last-millennium simulations from HadCM3 that indicated
significant reduction in global mean precipitation following these events. In the tropics, areas experiencing post-eruption drying



coincide with climatologically wet regions, while dry regions get wetter on average, but the changes are spatially heterogeneous. This pattern is of opposite sign to, but physically consistent with, projections under global warming.

Despite earlier studies, substantial uncertainties still affect our understanding of hydroclimate responses to volcanic forcing in certain tropical regions: for instance, the CMIP5/PMIP3 last-millennium ensemble does not robustly feature the

persistent drying over Mesoamerica that was detected in a recent speleothem record and ascribed to volcanically-induced changes in dominant modes of variability in the tropical Pacific and Atlantic, including ENSO (Winter et al., 2015). There are still apparent discrepancies in proxy-derived hydroclimate responses to volcanic eruptions in Asia and those simulated by several models (e.g. Anchukaitis et al., 2010) although it is not clear whether this is due to poorly simulated dynamics (and, if so, what dynamics are not adequately treated), uncertainties in forcing estimates, large contributions from internal variability, or

in the interpretation of the proxy network (Stevenson et al., 2016).

Among the known uncertainties, the hemispheric structure of stratospheric aerosol forcing plays a critical role in the hydroclimate response to volcanic events. Colose et al. (2016b) demonstrated that the response of the Intertropical Convergence Zone (ITCZ) differs dramatically for aerosol loading centered in the Northern vs. Southern Hemispheres in the CESM Last Millennium Ensemble (LME) and NASA ModelE2-R last-millennium simulations (Table 1), with the ITCZ shifting

away from the hemisphere with the greater concentration of aerosols (Haywood et al., 2013). Liu et al. (2016) also classified volcanic eruptions based on their meridional aerosol distributions, and noted that NH volcanic eruptions are more efficient in reducing NH monsoon precipitation than SH eruptions. Figure 6a characterizes precipitation anomalies after volcanic eruptions, while Figures 6b and 6d separate the precipitation response to volcanic forcing of differing meridional structure (18 ensemble members, see Colose et al. (2016a) for details on eruption classifications) in CESM LME during boreal winter in the Amazon

and boreal summer in the Sahel, respectively. Although there is considerable spread within a given classification related to different eruption magnitudes and internal variability, there is a strong tendency for an increase in local precipitation if the aerosol loading is preferentially located in the opposite hemisphere. Unique local responses to different volcanic eruptions, even for a particular configuration of the atmospheric state, therefore should not be expected. Figures 6c and 6e show the same result using all eruptions within three ensemble members in NASA GISS ModelE2-R. Volcanic eruptions were incorrectly

implemented in these three simulations (those using Gao et al., 2008) such that the aerosol optical depth for all events is approximately two times too large (Masson-Delmotte et al., 2013), but the implementation fortuitously allowed for an analysis of events with a very large signal.

In addition to the spatial distribution of volcanic aerosols, descriptions of volcanic aerosol cloud properties are also critical for simulations of the hydroclimate response to past eruptions. Substantial differences exist between reference volcanic

forcing datasets in terms of overall aerosol loading and their latitudinal structure (Gao et al., 2008; Crowley and Unterman, 2013; Sigl et al., 2015). The differences include the timing, magnitude, and spatial structure of the forcing, as well as the reported variable (e.g., AOD or aerosol mass). Furthermore, implementation of a given forcing dataset may differ among modeling groups, for instance by varying the aerosol effective radius (Crowley and Unterman, 2013; Zanchettin et al., 2016). Other understudied aspects of uncertainties in eruption characteristics may also play a substantial role in hydroclimate impacts,

such as plume composition, vertical profiles (LeGrande et al., 2016), and the start month of eruptions, the latter of which is often assumed constant but has a significant controlling influence on the structure of forced hydroclimate anomalies (Stevenson et al., 2017). A consideration of the forcing implementation is therefore critical for meaningful proxy-model comparisons and adds a dimension of uncertainty that is independent of model skill or the climatic interpretation of the proxies of interest.

The robustness of the ENSO response following volcanic eruptions has been a longstanding subject of inquiry, and

represents another important dynamic response that needs to be better understood in the context of hydroclimatic responses to





volcanic events. Many studies have yielded equivocal results, with some arguing that eruptions bias the tropical hydroclimate state toward El-Niño conditions (e.g. Mann et al., 2005; Emile-Geay et al., 2008; Li et al., 2013; Wahl et al., 2014; Maher et al., 2015), toward La-Niña conditions (D'Arrigo et al., 2008; D'Arrigo et al., 2009; McGregor and Timmermann, 2011; Zanchettin et al., 2012), or that there is no evidence of an ENSO response to volcanic forcing (Robock and Mao, 1995; Self et al., 1997;

Ding et al., 2014b; Tierney et al., 2015a). More recently, Pausata et al. (2016) have argued that eruptions with preferential aerosol loading in the NH will generate El-Niño conditions by virtue of a southward ITCZ shift, which slackens the trade winds over the Pacific Ocean. Stevenson et al. (2016) used the CESM LME to investigate the separate influences of volcanic eruptions and ENSO on hydroclimate variability. Hydroclimate anomalies in monsoon Asia and the western United States resemble the El-Niño teleconnection pattern following volcanic forcing that is either symmetric about the equator or

preferentially located in the NH. El-Niño events following an eruption can then intensify the ENSO-neutral hydroclimate signature. This implies that uncertainties in either the ENSO response to eruptions or the hemispheric loading of aerosols can contribute to proxy-model disagreement in hydroclimatic responses subsequent to eruptions. Multiple mechanisms, however, may be responsible for initiating this response (McGregor and Timmermann, 2011; Pausata et al., 2016; Stevenson et al., 2017). Internal variability in the form of a coincidental superposition of El-Niño events with volcanic eruptions can also be responsible

for proxy-model discrepancies (Lehner et al., 2016).

*3.3 Diagnosing Mechanisms of Hydroclimate Variability*

Although much progress has been made towards understanding the mechanisms of hydroclimate variability on annual and longer timescales, substantial uncertainties remain, particularly on multidecadal to centennial timescales. The linkages between

modes of coupled atmosphere-ocean variability and hydroclimate have been extensively explored on annual to decadal timescales, and also have been regularly exploited during the creation of proxy reconstructions of modes of atmosphere-ocean variability (Mann et al., 2009; Emile-Geay et al., 2013b; Li et al., 2013). Impacts from modes such as ENSO, the Atlantic Multidecadal Oscillation (AMO), the Pacific Decadal Oscillation (PDO) and the North Atlantic Oscillation (NAO) take several forms. Hydroclimate anomalies can result from Rossby wave-driven teleconnections, as in the ENSO influence on the Pacific-

North American and Pacific-South American patterns (Ropelewski and Halpert, 1986; Renwick and Wallace, 1996; Garreaud and Battisti, 1999); they can arise from changes to monsoonal flows, such as the AMO impact on North America (Oglesby et al., 2012); from alterations in the zonal-mean circulation in response to changes in ocean conditions during strong ENSO events (Seager, 2007); or in response to systematic shifts in the NAO following strong volcanic eruptions (Shindell et al., 2004; Ortega et al., 2015). This implies that the representation of coupled climate variability is critical for simulating hydroclimate variations

in climate models.

The precise contribution of different modes of variability to hydroclimatic changes is still an ongoing area of research. For the American Southwest, substantial work has indicated that interactions between modes such as ENSO, the AMO, and the PDO can affect the patterns and occurrence of megadrought over the CE (Coats et al., 2016b). The degree to which externally forced changes in coupled modes of variability modulate hydroclimate variations is another key target for the community. For

instance, ensemble simulations indicate that the overall forced changes to ENSO are minimal over the CE, but that volcanic activity substantially enhances the power in the AMO (Otto-Bliesner et al., 2016).

The availability of large model ensembles has demonstrated the key importance of internally coupled variability relative to the forced response in hydroclimate. Even for relatively strong external forcings, such as the 1815 eruption of Mt. Tambora, internal variability appears to be significant: the CESM LME shows an El-Niño event (and associated hydroclimate

impacts) following this eruption in only half of the ensemble members (Otto-Bliesner et al., 2016). Analyses of the CMIP5



simulations show similar scatter across ensemble members and across different models for 20[th]-century eruptions (Maher et al., 2015).

The coupled nature of the hydroclimate problem provides a unique opportunity to perform attribution studies using climate model simulations, and to quantify the sensitivity of hydroclimate to particular dynamical mechanisms in a manner impossible using paleoclimate reconstructions or observations alone. For example, Cook et al. (2013a) used experiments with and without prognostic dust aerosol physics to demonstrate the importance of dust mobilization to the persistence of megadroughts over the midwestern United States. More recently, Stevenson et al. (2015b) performed simulations in both a coupled configuration and in an atmosphere-only setup using a repeating 12-month SST climatology to show that internal atmospheric variability can generate megadrought-like behavior in the absence of coupled climate variations. Future targeted simulations will likely provide additional insights into the relevant dynamics associated with hydroclimate variability over the CE.

### 3.4 Opportunities for Modeling Progress

Climate simulations are affected by large systematic biases, including discrepancies between observed and simulated characteristics of both the mean state, seasonal cycles and variability (e.g. Flato et al., 2013; Wang et al., 2014a). The possible impacts of model biases in last-millennium simulations have been explicitly discussed only for a few specific regional aspects of climate dynamics and variability (e.g. Anchukaitis et al., 2010; Coats et al., 2013a; Zanchettin et al., 2015; Coats et al., 2016a; Stevenson et al., 2016). A broader assessment of model biases in simulated patterns, variability and teleconnections therefore would increase our confidence on both climate dynamics and variability inferred from climate simulations of the last millennium.

In some cases, bias is almost completely determined by lack of spatial model resolution and poorly resolved topography or bathymetry (e.g. Milinski et al., 2016). Increases in horizontal and vertical model resolution have the potential to not only improve the spatial detail of simulations, but also strongly alleviate or even remove some of the biases affecting the current generation of models. Another aspect of model improvement is increasing complexity, i.e., including an explicit or implicit treatment of additional processes such as cloud microphysics, cold pools and mesoscale organization of tropical convection. Progress in understanding hydroclimate variability is expected from the inclusion of water isotope tracers, so that isotopic fields can be calculated directly and compared with isotope-based proxy records, rather than making *a priori* assumptions about the "amount effect" or other factors controlling precipitation $\delta^{18}O$ and $\delta D$ (see Section 2). Additional progress is expected from an improved treatment of the indirect effects of aerosols.

In addition to model improvements, progress is expected through improved characterization and implementation of the external forcings used in last-millennium simulations. Forcing characteristics must be reconstructed based on indirect evidence, implying large uncertainties in the climatically relevant parameters of the forcing. For volcanic eruptions, for instance, this concerns the uncertainties discussed in Section 3.2. A recently published reconstruction of volcanic forcing (Sigl et al., 2015) has demonstrated advances in terms of more accurate timing of the events and a better estimate of the amplitude of sulfur emissions. Together with an improved module for translating volcanic emissions into aerosol optical properties (Toohey et al., 2016), the better constrained volcanic forcing reconstructions will lead to improved direct radiative and indirect dynamical responses in PMIP4 (Jungclaus et al., 2016). Further progress is expected from increased complexity of the forcing-related processes explicitly treated in the models. For instance, inclusion of modules for chemistry and aerosol microphysics improves representation of volcanic plume development and hence the climatic impacts of volcanic eruptions (LeGrande et al., 2016). Nonetheless, a multi-model assessment of the volcanic forcing generated by different aerosol climate models for a Tambora-like





equatorial eruption produces a substantial ensemble spread, raising questions as to the level of model complexity that is necessary to estimate the range of inherent climate uncertainty (Zanchettin et al., 2016). New reconstructions of solar irradiance also have been developed and progress is expected from a more consistent representation of spectral solar irradiance in the PMIP4 solar forcing data sets (Jungclaus et al., 2016). Improvements in atmospheric chemistry (particularly stratospheric and 715 tropospheric ozone responses) also increase the magnitude of response to solar changes (Shindell et al., 2006).

Finally, the complexity of current climate models implies numerous potential sources of uncertainty, the individual impact of which can be hard to distinguish in transient climate simulations. This establishes the need for coordinated modeling efforts to tackle specific aspects of uncertainty. For instance, the "Model Intercomparison Project on the climatic response to Volcanic forcing" (VolMIP; Zanchettin et al., 2016) has defined a set of idealized volcanic-perturbation experiments with well-720 constrained applied radiative forcing. The effort will identify primary limitations in model abilities to simulate climate responses to volcanic forcing, especially concerning differences in model treatment of relevant physical processes.

## 4. Comparing Proxy Data and Climate Model Simulations during the Common Era

Hydroclimatic changes during the CE provide a salient proving ground to investigate responses to major climatic forcings (e.g.,
solar, volcanic, greenhouse gases, land-use/land-cover changes) and also to better quantify the internal variability of the climate system. Such investigations are arguably best carried out leveraging the abilities of both climate model simulations and proxy data. Nevertheless, conducting meaningful, quantitative comparisons between proxies and models, or interpreting model and proxy information in tandem, is not straightforward. We therefore offer a set of "best practices" for integrating proxies and model simulations in the following subsections, all of which are aimed at improving comparisons of hydroclimate
reconstructions with model simulations and therefore enhancing our ability to both simulate climates of the past and future, and improving our interpretations of the signals recorded by proxies (e.g. Yoshimori et al., 2006).

### 4.1 Best Practices for Proxy-Model Comparisons of Hydroclimate

Proxy-model comparisons are most effective when they leverage the complementary strengths of both information sources. It is
tempting to view proxy-based data and reconstructions as the "ground-truth" against which model simulations should be judged. Paleoclimate data, however, are incomplete climate system indicators and this imposes various constraints and uncertainties that differ across proxies, many of which are summarized in Section 2. In the context of a climate reconstruction, these constraints can affect (among other things) the seasonality of the reconstructed climate signal, the spatial and temporal coverage and resolution of the reconstruction, and with what fidelity different timescales of variability (e.g., annual, decadal, centennial) are
recovered. As described earlier with regard to sediments, for instance, low frequencies in reconstructions of closed lake basins can arise simply by the integration of high-frequency climatic signals (Huybers et al., 2016), while tree-rings, because of their limited length, may not accurately represent long-term (multicentennial) variability.

Climate models, in turn, can provide nearly complete information on any climate system variable or process of interest (e.g., temperature, precipitation, circulation) for any time interval in a manner internally consistent with model physics. They
may, however, diverge from the "true" state of the climate system because of specific physical processes that are either not represented or contain representational errors, misdiagnoses of climate forcings, or mismatches in the timing or spatial pattern of internally generated variability, as discussed in Section 3. Proxy-model comparisons are therefore best cast in a framework in which simulations and proxy observations compensate for the deficiencies that they each contain. Given these general considerations, we articulate below several recommendations for hydroclimatic proxy-model comparisons:




1. *Expectations of temporal or spatial consistency between proxies and models should be critically evaluated.*

Given natural variability in the climate system, it is unrealistic to expect congruent time histories of hydroclimate derived from proxy data and model simulations. Direct time comparisons can be misleading, especially if events in the paleoclimate record are associated with internal variability rather than exogenous forcing. Internal (unforced) variations will evolve differently from one model simulation to the next if initial conditions are not identical, and amplitudes of decadal and longer timescale modes of climate variability (e.g., the AMO, PDO) differ between model frameworks (Deser et al., 2012; Zanchettin et al., 2013; Kay et al., 2015; Fleming and Anchukaitis, 2016; Otto-Bliesner et al., 2016). As an example, Figure 7 shows the PDO index (Phillips et al., 2014) from four full-forcing ensemble members of the LME (Otto-Bliesner et al., 2016). Each run is forced identically, but started from slightly different atmospheric initial conditions, resulting in unforced patterns of variability (e.g., the PDO) that are uncorrelated with one another over the majority of the simulation interval (Kay et al., 2015; Fleming and Anchukaitis, 2016).

Despite the uncorrelated temporal behavior, however, the power spectra of these time series (Figure 7) are quite similar to each other in their underlying distributions. This highlights the fact that although temporal correlation is not expected, proxy-model comparisons are still meaningful as long as they focus on the underlying basic processes and full-spectrum of variability. Such comparisons may include statistics of extremes (e.g., droughts), analogues (e.g., similar events in models and reconstructions that share certain characteristics but do not necessarily align in time), and physical mechanisms (e.g., the teleconnection between ENSO and drought in a given region). For example, Coats et al. (2015b) used a suite of PMIP3/CMIP5 simulations to investigate ocean forcing of North American megadroughts during the Medieval era. In this case, the models did not match the timing of the megadroughts seen in the proxy reconstruction, but were able to produce all other megadrought aspects (e.g. frequency, duration, etc.) through variability and teleconnections that were in some cases consistent with observations. Similar conclusions are evident in Figure 8, which compares the occurrence of megadroughts as estimated by the North American Drought Atlas (NADA; Cook et al., 2004), the PMIP3 ensemble of PDSI, and PDSI calculated from the LME.

Models may also misrepresent the exact spatial signature of climate phenomena, such as the spatial extent of monsoon regions or coupled modes of hydroclimate variability such as the ENSO or the Indian Ocean Dipole. Proxy-model comparisons can proceed, however, as long as the spatial representation in the model is taken into consideration when comparing with proxy data (e.g. Konecky et al., 2014), similar to assessments with instrumental data (Marvel and Bonfils, 2013).

2. *Proxy-model comparisons should be viewed as a "two-way conversation."*

Proxy-model comparisons are most effective when viewed as a "two-way conversation" between two independent sources of information that can be used to propose and test process-based hypotheses. For example, model simulations can be *a priori* designed to better understand outstanding features in the CE paleoclimate record, such as megadroughts and volcanic-related coolings (e.g. Seager et al., 2005; Stevenson et al., 2015b). These may include, for example, single-forcing simulations that can isolate mechanisms that may explain an observed climatic response (e.g. Schurer et al., 2013; Otto-Bliesner et al., 2016). On the other hand, features that emerge in climate model simulations can serve as targets for paleoclimatic investigation. For example, Tierney et al. (2015b) observed that future climate model simulations overwhelmingly show that East Africa will get wetter in the 21[st] century under rising greenhouse gases as a result of a simulated "El-Niño-like" mean state in the tropical Indo-Pacific. This motivated the





construction of a hydroclimate record from East Africa to test the relationship between regional precipitation, recent warm climate intervals, and the mean state of the Indo-Pacific system. The paleoclimate record showed that the 20[th] century was actually dominated by a strong drying trend unprecedented in the last millennium, indicating that greenhouse gases may in fact be driving the East African region toward a drier state; the wetter prediction of the

models associated with greenhouse gas forcing therefore deserves further investigation.

3. *Uncertainties and limitations of models and proxies must be rigorously assessed to achieve robust conclusions from proxy-model comparisons.*

       The comparison of paleoclimate reconstructions and models will always be challenged by the presence of

uncertainties inherent to each data source. The magnitude of this uncertainty differs depending on the application and such uncertainties are largely unavoidable. Nevertheless, there are ways to limit the effect that this uncertainty has on subsequent hydroclimatic comparisons and interpretations.

       Summarizing from Section 3, the major sources of uncertainty for climate models are: 1) limits to spatial resolution, causing errors in the simulation of topographical effects and finer-scale processes; 2) parameterization of

physical processes (e.g. small thunderstorms cannot be directly modeled); 3) tuning to modern-day climate (i.e. adjusting model parameters to "fit" the observations of the present day); and 4) selection of forcing data. External users should always ensure that the model data selected are suitable for their purpose and should be aware of the limitations associated with factors 1-4. Important means of quantifying the sensitivity of proxy-model comparisons to these sources of uncertainty include assessment of structural modeling error (Dee et al., 2016) and the use of multi-

model ensembles (Coats et al., 2015a; Coats et al., 2015b; Coats et al., 2015c; Lewis and LeGrande, 2015; McGregor et al., 2015) to characterize structural uncertainty, evaluating model simulations of the relevant processes, and using an ensemble of simulations (Otto-Bliesner et al., 2016) to address internal variability.

       For hydroclimate comparisons, some of the sources of uncertainty can be minimized. Precipitation amount is a complex target for proxy-model comparisons due to highly localized influences such as topography, which can skew

the precipitation regime at any individual location. Moreover, the coarse nature of model grids means that a single grid point may represent an area on the order of 100 km$^2$ or larger. Comparing station observations and model grid-boxes is therefore problematic due to contrasting geographic resolutions (e.g. Osborn and Hulme, 1997) and the same is true for proxy data that represent only a small area or a single point in space. This precludes making direct comparisons between a reconstruction from a single location, for example from an individual cave or lake, and the nearest grid cell

in a model. Directly comparing gridded model output to data at a specific point therefore should be avoided, and while such a comparison is tempting, in most cases it is either falsely analogous, for instance some models are not expected to reproduce highly regional hydroclimate dynamics (e.g. orographic rainfall or mountainous microclimates), or unrealistic, for instance CMIP5-class models underrepresent or omit small landmasses (Karnauskas et al., 2016), such as the grid boxes nearest the Perida Cave, Puerto Rico reconstruction (Winter et al., 2011) that are represented as ocean

in the PMIP3 simulations (not shown).

       To avoid the potentially large uncertainties associated with the coarse spatial resolution of climate models, proxy-model comparisons should focus on atmosphere-ocean dynamics or hydroclimate features with large spatial scales that span multiple grid boxes. This is usually on the order of thousands of kilometers or larger. For analogous comparisons, where multiple proxy records are available, a synthesis of these records provides better estimates of the

region's hydroclimate variability than a single record, and is more suitable for comparisons with the spatial patterns





simulated by models (e.g. Tierney et al., 2013; Otto-Bliesner et al., 2014; Cook et al., 2016). If comparing a hydroclimate reconstruction from a single location with a model simulation is absolutely necessary, the model variable of interest should be considered over a regional scale to account for the possibility that the simulated hydroclimate dynamics do not exactly covary with the location of the reconstruction (that is to say, the nearest model grid point may not be the most relevant given the model physics). Alternatively, regional climate simulations, produced from dynamically or statistically downscaled products, should also be considered for comparison, although such downscaling introduces additional structural uncertainties.

Summarizing from Section 2, some of the major uncertainties from proxy archives include: 1) variables measured in the proxy archive may not match a variable available from the model; 2) inherent non-climatic effects decrease the signal-to-noise ratio; 3) spectral reddening; and 4) chronological error. As discussed above, the latter may be mitigated as long as the focus of the proxy-model comparison is process-based and not tied to the identification of specific events at specific times. Steps to mitigate Factor 1 are discussed in the subsequent subsection. Factor 2 can be partially addressed by designing optimal proxy networks that target climatically-sensitive regions (Evans et al., 1998, 2002; Comboul et al., 2015; Lewis and LeGrande, 2015; Dee et al., 2016) and statistical pre-treatment; see for example the discussion regarding isolation of the climatic signal in tree rings in Section 2.2.

The issue of spectral reddening (Factor 3) deserves a special mention, given that several previous studies have compared low-frequency variability in proxy records and model simulations (Ault et al., 2013a; Ault et al., 2013b; Bunde et al., 2013; Franke et al., 2013; Markonis and Koutsoyiannis, 2016). It should first be noted that these efforts are dependent on the employed proxy archives and may involve incompatible comparisons between model output and what proxies actually record (c.f. discussion of Franke et al. (2013) in Section 2.2 and Figure 2). Most proxy archives also contain at least some non-climate sources of autocorrelation that results in reddening of the primary climatic signal. As discussed in Section 2, trees store carbohydrates in their needles, leaves, and cambium (Fritts, 1976), trees suffer injuries due to short-term climatic events (e.g., droughts) that may affect growth rates for years after the event (Anderegg et al., 2015), soil moisture persists from one season to the next (Delworth and Manabe, 1993), and water in lakes and caves can take years or decades to cycle through such archives (Hurst, 1951; Benson et al., 2003; Evans et al., 2006; Truebe et al., 2010). Accordingly, spectral reddening will distort the importance of low-frequency variations by making them seem more substantial in the proxy than they are in the underlying climate system (c.f. Figure 3)

Misleading conclusions regarding the spectral power of the climate system can be avoided by: 1) choosing an appropriate quantity from model simulations to compare to proxy-based reconstructions that accounts for autocorrelation; and 2) applying a forward model of proxy behavior (see following recommendation) to model output to explicitly simulate spectral reddening. When these steps are taken, spectral comparisons between models and proxies can be undertaken with confidence; model PDSI (or soil moisture) and tree-ring reconstructed PDSI offer an example of a successfully analogous comparison (Figure 2). Comparing proxies and models on "equal ground" indeed provides a tractable way to deal with proxy- and model-related uncertainties that has the potential to significantly advance proxy-model comparisons, as highlighted in the next recommendation.

4. *Proxy-model comparisons should take place on "equal ground"*

As highlighted in Section 2, data that can be measured in proxy archives are in many cases indirect indicators of hydroclimate rather than concrete variables like precipitation or evaporation that are the standard output for a climate model. To reconcile these different types of information and facilitate meaningful comparisons, attempts should be







made to either transform the proxy into a climatic variable, or transform climate model output into a proxy variable. Conversion of the proxy into a climatic variable can be achieved through regression or traditional inverse methods as is done with the widely used regional drought atlases (Cook et al., 2004; Cook et al., 2010a; Cook et al., 2015b; Palmer et al., 2015). Nevertheless, in many cases the multivariate nature of proxy data, the presence of large uncertainties and

limited spatiotemporal coverage in a calibration proxy network, or non-stationary behavior between the proxy predictor and the climate predictand, render regression and inversion challenging (e.g. Wilson et al., 2010; Lehner et al., 2012; Smerdon, 2012; Tingley et al., 2012; Coats et al., 2013a; Gallant et al., 2013; Evans et al., 2014; Konecky et al., 2014; Raible et al., 2014; Wang et al., 2014b; Wang et al., 2015; Konecky et al., 2016).

In many situations, conversion of model data into a proxy variable is more tractable, and is increasingly being

applied. Proxy System Models (PSMs; Evans et al., 2013) that mathematically describe the way in which climate is encoded into a proxy archive provide an ideal method of translation. PSMs are simplified representations of a biological, physical or chemical sensor's response to environmental processes, which is subsequently imprinted onto a physical archive and observed via physical or geochemical measurements (Evans et al., 2013; Dee et al., 2015). PSMs are forward models and therefore do not require assumptions of local/regional stationarity and linearity that are

typically applied in inverse approaches, and they can characterize multivariate influences (Evans et al., 2013; Phipps et al., 2013). Paleoclimate observations in their "native" units may be directly compared to models by using the output variables of simulations or other climate models that have been processed through a PSM to generate simulated paleoclimate observations – sometimes called "pseudoproxies" in the literature (Thompson et al., 2011; Tolwinski-Ward et al., 2011; Smerdon, 2012; Phipps et al., 2013; Evans et al., 2014). PSMs may therefore provide more

appropriately analogous comparisons for models and proxies; they also allow for the propagation of uncertainties because PSM-derived estimates of parameter errors and observational random error do not need to be inverted. It is important to note, however, that PSMs are simplified representations of the proxy system and do not perfectly capture all the vagaries of actual proxy data. In many cases, limitations in our mechanistic understanding of the proxy sensor or archive, or the existence of processes that are mathematically complex, preclude the development of a truly process-

based model. For example, Thompson et al. (2011) developed a PSM for coral $\delta^{18}O$ based on geochemical expectations rather than explicitly modeling fractionation during the calcification process, the details of which are understudied. Further refining PSMs is therefore an important area of research, as is testing PSM results with observational data

If there is one particular quantity for which establishing "equal ground" comparisons is paramount for our

understanding of hydroclimate, it is the isotopes of water (oxygen and hydrogen). Water isotopes are measured in every major paleoclimatic archive (trees, speleothems, corals, sediments, ice cores) and are complex but powerful indicators of the hydrological cycle. Isotopic records reflect large-scale atmospheric processes, like changes in precipitation source region, vapor transport, local amount effects, and the seasonality of precipitation, and often capture large-scale climate modes such as ENSO and the Indian Ocean Dipole, as well as monsoon characteristics, more

strongly than precipitation amount at a given location (Vuille et al., 2005a; Vuille et al., 2005b; Vuille and Werner, 2005; Konecky et al., 2014; Moerman et al., 2013; Moerman et al., 2014). Water isotopes are therefore in many ways ideal indicators for comparison with coarse-resolution model output (Jouzel et al., 1994; Hoffmann et al., 1998; Schmidt et al., 2007; Risi et al., 2010; Pausata et al., 2011; Dee et al., 2015). The best way to analyze isotope proxy information is to compare data to climate models with isotopes incorporated into the simulation. Recent simulations

that include isotope tracers have greatly simplified the task of undertaking proxy-model comparisons for a large





component of the proxy data that are available during the CE (Colose et al., 2016a). Intercomparison projects like the SWING2 project are also compiling the results from isotope-equipped atmospheric climate models (Sturm et al., 2010; Conroy et al., 2013) to provide the community with a diverse set of simulations to probe the drivers of spatiotemporal water isotope variability. Further efforts to even more widely include isotope tracers in coupled climate models are

nevertheless needed and would significantly advance our ability to compare proxy and model simulations.

*4.2 Model-Data Comparisons with Data Assimilation*

An emerging, promising and specific means of proxy-model comparison that can incorporate the use of water isotope modeling and PSMs is data assimilation (DA). DA for paleoclimate optimally fuses proxy information with the dynamical constraints of

climate models (Goosse et al., 2010; Widmann et al., 2010; Goosse et al., 2012b; Steiger et al., 2014; Hakim et al., 2016; Matsikaris et al., 2016). Unlike traditional statistical reconstruction approaches, DA-based reconstructions simultaneously produce hydroclimate and dynamically related climate variables that can be used to assess the physical causes of past climate changes. Another key advantage of the approach is that it incorporates additional information into a fundamentally information-poor problem: DA-based reconstructions have been able to achieve high reconstruction skill (Hakim et al., 2016) by merging

model covariance information with noisy proxy data. Such reconstructions specifically targeting hydroclimatic fields are currently being developed, and the approach is expected to provide a more complete picture of hydroclimate over the past millennium, while allowing tests of many existing hypotheses about hydroclimatic change in the past. For instance, many dynamics-based hypotheses have been proposed to explain periods of anomalous hydroclimate, such as the apparent clustering of megadroughts in the American Southwest during the Medieval period (Graham et al., 2007; Herweijer et al., 2007; Seager et

al., 2008; Coats et al., 2016b). DA-based hydroclimate reconstructions may be able to more clearly elucidate the dynamics associated with droughts, pluvials, their temporal phasing across multiple regions, and their multidecadal-to-centennial variability over the past millennium on a truly global scale because the method simultaneously provides additional dynamically-relevant fields in addition to reconstructions of specific hydrocimate variables (as opposed to more traditional approaches that may target dynamical fields or indices independently).

While DA methods provide many opportunities, we note two major challenges. The foremost challenge is the high computational cost: for an ensemble DA scheme, cycling tens or hundreds of CMIP5-class coupled climate models for a thousand years or more is simply impossible given the limits on current technologies. Two main approaches have emerged in response to this issue, either using computationally efficient "intermediate complexity" models with small ensemble sizes (e.g. Goosse et al., 2010; Goosse et al., 2012b) or using an "off-line" DA approach where large ensembles of climatologically

plausible states are drawn from pre-existing climate simulations (e.g. Steiger et al., 2014; Hakim et al., 2016). Another major challenge is the integration of proxy data, PSMs, and climate models, each of which must work in concert while being subject to their own uncertainties and assumptions (Dee et al., 2016). It is therefore critical that such DA-based reconstructions be well validated and account for uncertainties in all of the employed information sources.

*4.3 Schematic Summary of Proxy-Model Comparisons*

The schematic presented in Figure 9 summarizes the challenges and recommendations regarding proxy-model comparison discussed in this section. These include practices for interpreting the data directly from climate models, measured proxy data, as well as data from PSMs and paleoclimate reconstructions. Note that the data produced by the PSMs and reconstructions have been transformed (e.g. via a statistical model) so that the values from the model and/or proxy data are directly comparable. The

red boxes in Figure 9 show the major sources of uncertainty that are implicit to each of the data sources. The arrows in Figure 9



show the usual paths of proxy-model comparisons and those steps requiring a transformation are shown in blue (e.g. from proxy data to paleoclimate data, or from model data to proxy data via a PSM).

**5. Quantitative estimates of future hydroclimatic risks**

A critical goal of proxy-model comparisons over the CE is to help assess future risks of hydroclimatic anomalies under global warming. In particular, extreme events and decadal-scale predictability are attractive targets for climate prediction, because knowledge of these features is useful for water resource managers, stakeholders and regional planners. On these timescales climate is likely to evolve as a consequence of both forced changes from human activity as well as internal climate fluctuations. Risks of extreme and/or persistent hydroclimatic phenomena (both pluvials and megadroughts) will therefore be sensitive to

both of these contributions. For example, if unforced decadal-to-multidecadal variations are strong (high amplitude), then they could mask long-term forced changes; estimating prolonged drought risk in such a setting would require models (either statistical or dynamical) of hydroclimatic variability to include comparably high-amplitude variability at these timescales to simulate possible near-term climate trajectories. If, on the other hand, decadal-to-multidecadal variability is weak, then regional hydroclimate will likely evolve primarily as a function of forced changes superimposed onto year-to-year variability. For

example, consider an area projected to get wetter as a consequence of anthropogenic climate change. If the actual decadal variability is more important to hydroclimate in this region than represented in climate models prolonged drought risk would be underestimated by model simulations. In contrast, if the amplitudes of model and proxy-based estimates of decadal-to-multidecadal variability are similar, confidence in near-term model projections will be enhanced. It is also desirable to quantify the long-term mean state of the system because limited instrumental observations could overlap in part with decadal-scale

anomalies not representative of the multi-centennial average.

The motivations and considerations above are critical and can be addressed by proxy-model comparisons over the CE because the magnitude of internal variability and the long-term mean state of hydroclimate can be more accurately estimated over the time period. The promise of improving these estimates is that the range of possible conditions in the future, and therefore the risk that we associate with a given hydroclimatic anomaly, will be better constrained. Below we review two

approaches that have used the paleoclimate record and model simulations to quantify future hydroclimate risks. The examples are relevant to water resources in the western United States, and in particular semi-arid regions with significant populations exposed to hydroclimate risks, but such applications are broadly applicable in regions where sufficient paleoclimate data are available.

*5.1 Probabilistic Estimates of Megadrought Risk*

Estimating the risk of decadal-scale phenomena such as megadroughts during the first half of this century can be cast in terms of the probability of a given indicator (e.g. a drought index, soil moisture or precipitation anomalies) falling below some reference threshold for an unusual amount of time. This probability can be written $\Pr\{X < q | \Delta\mu, \sigma'\}d$, where $X$ is the time series of interest – for instance, PDSI – averaged over a sufficiently long period of time, $q$ is a threshold for megadrought, and $\Delta\mu$ and $\sigma'$

are changes in the mean and standard deviation of $X$ relative to some historical baseline (Ault et al., 2014; Cook et al., 2014; Ault et al., 2016; see also Figure 8). One means of estimating $\Pr\{X < q | \Delta\mu, \sigma'\}$ is accomplished by deriving the parameters describing $\Delta\mu$ and $\sigma'$ directly from ensembles of climate model simulations (Ault et al., 2014; Cook et al., 2015a). These parameters are then used to generate many realizations of Monte Carlo time series that emulate the shifted mean and modified standard deviation of $X$. The occurrence rate of megadroughts in such realizations is calculated from each constituent model

simulation or ensemble member. Risk is therefore expressed as the fraction of all realizations that include at least one





megadrought for a given change in mean and variability and all estimates from all realizations can be averaged to generate a "master" multi-model probability density function (PDF). Alternatively, megadrought probabilities can be assessed by computing risks over a plausible range of values for the mean and standard deviation to generate a two-dimensional PDF. Megadrought risks estimated from specific models can then be assessed by identifying where in the two-dimensional PDF they

simulate future hydroclimate. This approach can also be used to identify the individual contributions of temperature and precipitation to megadrought probabilities (Ault et al., 2016).

While the above approaches can be pursued exclusively with climate model simulations, paleoclimate information can be incorporated and used to enhance the characterization of future changes in hydroclimate. Specifically, the spectral characteristics of hydroclimate in a given region (e.g., *X* from above) can be quantified, and then used to constrain the frequency

characteristics of the Monte Carlo time series used to emulate *X*. For example, Ault et al. (2014) used paleoclimate estimates of the continuum of hydroclimate variability in the southwestern United States to inform the stochastic noise generators used in the Monte Carlo procedure. This approach is based on the argument that last-millennium hydroclimate variability is generally higher in proxies than in models, but it remains unclear if this difference reflects true deficiencies in models, their forcings, non-climate processes responsible for modifying proxy spectra (Section 2), or some combination of these factors. Moreover, this

approach has not been widely or systematically applied to other parts of the world, let alone to near-term time horizons. PSMs (Section 4) also offer an especially useful opportunity for reconciling model and proxy spectra that could also be used in these contexts, but they have not yet been applied to help constrain decadal-to-multidecadal hydroclimate amplitudes.

The quantitative megadrought risk estimates discussed above are best suited to the second half of the 21[st] century, when forced changes dominate the mean state and may also affect hydroclimatic variability. Estimating megadrought hazards

during the coming decades requires that initial conditions be factored into any credible assessment. One strategy to do so would be to use decadal prediction (Meehl et al., 2009) to estimate near term values of $\Delta\mu$ and $\sigma'$, then apply the direct method described above to complete a PDF of megadrought risk. Paleoclimate constraints on the amplitude of decadal to multidecadal variations in the Monte Carlo time series could likewise be applied, yielding raw and "paleoclimate-informed" estimates of risk.

In all of the above applications, several guiding principles can be articulated for future work that involves proxy-model

comparisons for the purpose of constraining probabilistic estimates of megadrought risks. First, prolonged hydroclimate events should be identified as anomalies in long-term (either decadal or multidecadal) averages with respect to an historical baseline that can be constrained by paleoclimatic estimates. Secondly, prolonged hydroclimate events are infrequent events, making Monte Carlo (or other statistical) methods useful for generating robust estimates of their probability of occurrence. At some future date, ensemble sizes may include hundreds of members, not just dozens (e.g. Kay et al., 2015), yielding yet another

method for estimating the PDF of hydroclimate risks (Coats and Mankin, 2016). In either of these approaches, the statistical characteristics of decadal-to-centennial variability in proxies will continue to provide important constraints on these timescales and the ability of models to realistically reproduce them.

### 5.2 Riverflow reconstructions applied to water resources management

Studies of riverflow represent another area in which proxy-model comparisons have been important for informing future risk projections. Long proxy reconstructions of riverflow constrain the historical mean state and range of variability in the amount of water flowing through major river systems, which are a critical resource particularly in arid and semi-arid environments. In the western United States, for instance, the water supply for states including California and Arizona depend on an allocation of the annual flow of the Colorado River (Pulwarty et al., 2005). Rising temperatures, changes in precipitation, and alterations to

snowfall amount and snowmelt timing threaten to reduce flow in the Colorado and other western North American rivers



(Christensen et al., 2004). In addition to concerns about on-going and future climate-change modification of riverflow, tree-ring reconstructions covering parts of the CE also suggest that past megadroughts were substantially more severe and prolonged than observed during the last century (Woodhouse et al., 2006; Meko et al., 2007; Gray et al., 2011) and that observations from the early 20th century used to allocate water from the Colorado River reflected an exceptionally wet period relative to the rest of the CE (Stockton and Jacoby, 1976). Paleoclimate estimates therefore provide more complete information for water managers and policymakers on the possible range of variability in interannual, decadal, and multidecadal flows of the Colorado River (Woodhouse and J., 2006; Woodhouse and Lukas, 2006; Rice et al., 2009; Meko and Woodhouse, 2011). Moreover, reconstructions from multiple river systems reveal the frequency of multi-basin droughts that could affect the large-scale water supply infrastructure across the western United States (Meko and Woodhouse, 2005; MacDonald et al., 2008). Riverflow reconstructions from tree-rings are also not limited to the western United States and can be developed in other basins where tree-ring proxies reflect large-scale seasonally-integrated moisture balance (e.g. Lara et al., 2008; Wils et al., 2010; Urrutia et al., 2011; Cook et al., 2013b; Allen et al., 2015).

The widespread development of riverflow reconstructions offers the ability to constrain model predictions of the future using these paleoclimatic estimates. Barnett and Pierce (2009) evaluated a range of future water delivery scenarios for the Colorado River, using tree-ring estimates of its mean riverflow in combination with imposed flow reductions intended to mimic possible future climatic changes. They found that using the long-term CE reconstructed mean flow resulted in an increasing likelihood of future shortfalls and unsustainable water deliveries in the 21st century, compared to using 20th century observations of riverflow alone, and that climate change further exacerbated these shortfalls. Meko et al. (2012) used a river management model to show that a re-occurrence of 12th-centuy Medieval megadroughts in the Colorado River Basin would reduce Lake Mead to 'deadpool,' the point at which water falls below the outlet at Hoover Dam, within a few decades. Combining modern flow observations with tree-ring reconstructions, they also estimated that a megadrought as severe as the 12th century would be expected to occur every 4 to 6 centuries, even in the absence of climate change (Meko et al., 2012). Miller et al. (2013) combined observations and tree-ring reconstructions of Gunnison River (Colorado) flow to predict the probability of future wet and dry regime shifts. Thus, paleoclimate reconstructions of riverflow not only more fully characterize the range of variability across timescales and reveal long-term changes in the mean state, but can also be integrated with water management tools and used in probabilistic analyses to provide quantitative information about future water availability in these systems.

The examples provided in this subsection and in Section 5.1 illustrate how proxies and models have been used to collectively constrain future hydroclimatic risks. The approaches have applicability in other parts of the world beyond western North America, as well as to other types of prolonged hydroclimate extremes beyond droughts. The utility of these efforts are therefore widespread and significant, and ultimately can provide stakeholders and water resource managers with new tools to prepare for a realistic range of near- and long-term climate change outcomes.

## 6. Conclusions

The focus of this contribution has been on hydroclimatic proxy-model comparisons over the CE. The challenge of such comparisons over this time interval is developing a set of practices to combine the individual strengths of proxies and models to synergistically reveal fundamental insights about the climate system that each would be unable to achieve alone, but to do so with appropriate deference to their numerous individual and combined limitations. Hydroclimate presents a unique challenge in this regard because precipitation is more variable in time and space than temperature, rendering characterization of its past and future behaviour a challenge in both proxies and models. Moreover, many hydroclimatically relevant variables like soil moisture, runoff, transpiration, and water isotopes are subject to multivariate influences that can be convoluted in both proxy



and model analyses. Despite these challenges, this review highlights the substantive advances in hydroclimate reconstructions and model simulations and their associated comparisons, as well as best practices for such efforts in the future. It complements previous workshop assessments that have focused on how proxy-model comparisons over multiple paleoclimatic intervals can be used to constrain future projections (Schmidt et al., 2014) and more recent work that investigates CE proxy-model

comparisons specifically for temperature (PAGES 2k-PMIP3 Group, 2015). We have reviewed the most relevant hydroclimatic proxies available for study over the CE (Section 2) and the state of the science regarding last-millennium simulations from fully-coupled climate models (Section 3). In each case, we have articulated contemporary interpretations of these data sources, the current understanding of their uncertainties and how each sub-field is moving forward to address them. Given the context of our assessment of available proxy and model information, we have emphasized a set of considerations and best practices for

proxy-model comparisons of hydroclimate over the CE (Section 4). The recommendations are as follows:

1. *Expectations of temporal or spatial consistency between proxies and models should be critically evaluated.*

2. *Proxy-model comparisons should be viewed as a "two-way conversation."*

3. *Uncertainties and limitations of models and proxies must be rigorously assessed to achieve robust conclusions from proxy-model comparisons*

4. *Proxy-model comparisons should take place on "equal ground"*

These recommendations can be generalized beyond proxy-model comparisons of hydroclimate during the CE, but also provide important and specific context for hydroclimate proxy-model comparisons over the CE at a time when such investigations are only just becoming possible. Finally, a particularly important potential application of paleoclimatic proxy-model comparisons is their use in quantitatively constraining risk projections of future climate change. We have reviewed initial efforts to achieve

these constraints specifically in the context of megadrought risks and riverflow estimates (Section 5). These examples represent starting points for applications that seek to explicitly use paleoclimatic information to evaluate models and the risk assessments associated with future emissions pathways that they provide. Such risk assessments are fundamentally based on the ability of climate models to simulate the full range of internal hydroclimate variability, in addition to their forced responses, and the CE is therefore a vital paleoclimatic target for assessing the ability of models to do so.


*Data availability.* The coral based reconstructions of seawater $\delta^{18}O$ ($\delta^{18}O_{sw}$) shown in Figure 1 were downloaded at PANGAEA and NOAA Paleoclimatology (Felis et al., 2009b; https://doi.pangaea.de/10.1594/PANGAEA.743956; Hetzinger et al., 2017; https://doi.pangaea.de/10.1594/PANGAEA.873994), and annual averages with respect to the calendar year were calculated where necessary (Nurhati et al., 2016; https://www.ncdc.noaa.gov/paleo-search/study/10373; Zinke et al., 2016;

https://www.ncdc.noaa.gov/paleo-search/study/8606; Linsley, 2017b; https://doi.pangaea.de/10.1594/PANGAEA.874078; Linsley, 2017a; https://doi.pangaea.de/10.1594/PANGAEA.874070). The coral Sr/Ca and $\delta^{18}O$ data needed to calculate the pentannual resolution $\delta^{18}O_{sw}$ reconstruction from the Great Barrier Reef shown in Figure 1 are available at NOAA Paleoclimatology (Hendy et al., 2016; https://www.ncdc.noaa.gov/paleo-search/study/1869) and the corresponding methods are reported in the original publication (Hendy et al., 2002). We are in contact with Hendy et al. and foresee the archiving of these

data in a public database if the paper advances to publication following the Discussion phase. Drought atlas and tree-ring data used in Figures 2 and 8 are publicly available through the National Centers for Environmental Information (https://www.ncdc.noaa.gov/data-access/paleoclimatology-data/datasets/tree-ring/north-american-drought-variability) and the International Tree-Ring Data Bank (https://www.ncdc.noaa.gov/data-access/paleoclimatology-data/datasets/tree-ring). Instrumental-based self calibrating PDSI data used in Figure 2 are available through the Climatic Research Unit of the

University of East Anglia: https://crudata.uea.ac.uk/cru/data/drought/#global. GPCC data used in Figure 2 are available through



the Earth System Research Laboratory (https://www.esrl.noaa.gov/psd/data/gridded/data.gpcc.html), as are the GPCP data used in Figure 5 (https://www.esrl.noaa.gov/psd/data/gridded/data.gpcp.html). Proxy data in Figure 3 are available through the National Centers for Environmental Information (https://www.ncdc.noaa.gov/paleo-search/study/13686), as are the GHCN station data (https://www.ncdc.noaa.gov/data-access/land-based-station-data/land-based-datasets/global-historical-climatology-network-ghcn). As noted in its caption, Figure 4 is a reproduction of Figure 7 in Ge et al. (2008b). PMIP3 and CMIP5 data used in Figures 5 and 8 are available through the Earth System Grid Federation: https://esgf.llnl.gov/. All CESM LME and LENS data used in Figures 6, 7, and 8 are available through the Earth System Grid at the National Center for Atmospheric Research: https://www.earthsystemgrid.org/home.html. Forcing data plotted in Figure 8 are available through the National Centers for Environmental Information: https://www.ncdc.noaa.gov/data-access/paleoclimatology-data/datasets/climate-forcing.

*Acknowledgements.* This paper reflects findings and discussions from a workshop held at the Lamont-Doherty Earth Observatory of Columbia University on June 1-3, 2016, titled *Comparing data and model estimates of hydroclimate variability and change over the Common Era* (see http://pages2kpmip3.github.io/ and Smerdon et al. (2016)). The PAGES2k and PMIP communities were instrumental in the motivation and conceptualization of the workshop and we gratefully acknowledge funding from PAGES and the Lamont Climate Center. We thank the World Climate Research Program's Working Group on Coupled Modeling, which oversees CMIP, and the individual model groups (listed in Table 1) for making their data available.

Individual researchers on this study were supported in part by the US National Science Foundation through the National Center for Atmospheric Research and grants AGS-1502150, AGS-1303976, AGS-1433551, AGS-1602564, AGS-1501856, AGS-1243125, AGS-1243107, AGS-1404003, AGS-1243204, AGS-1401400, AGS-1602581, AGS-1502830, AGS-1203785 and OCE-1502832; Australian Research Council grants SR140300001, FL100100195 and DECRA DE16010092; National Natural Science Foundation of China under grant numbers 41305069 and 41675082 and the Jiangsu Collaborative Innovation Center for Climate Change; the German Federal Ministry of Education and Research (BMBF) and JPI-Climate/Belmont Forum projects PACMEDY (FKZ: 01LP1607B) and Collaborative Research Action "INTEGRATE, An integrated data-model study of interactions between tropical monsoons and extratropical climate variability and extremes;" the DFG-Research Center/Cluster of Excellence 'The Ocean in the Earth System' at the University of Bremen; the Swedish Science Council (VR) (2012-5246); the Brown Institute for Environment and Society; the Lamont-Doherty Postdoctoral Fellowship and Earth Institute Postdoctoral Fellowship programs of Columbia University; and the NOAA Climate and Global Change Postdoctoral Fellowship Program, administered by the University Corporation for Atmospheric Research Visiting Scientist Program. All authors contributed to the conception and scope of the paper during the course of the aforementioned workshop and in subsequent exchanges. J. Smerdon, J. Luterbacher and S. Phipps coordinated and led the construction of the manuscript. Sections 1 and 6 were jointly written by all authors. Principal contributions for each technical section were composed by the writing team as follows (in alphabetical order): Sect. 2.1: K. Cobb and T. Felis; Sect. 2.2: K. Anchukaitis and J. Maxwell; Sect. 2.3: K. Cobb and J. Partin; Sect. 2.4: J. Tierney; Sect. 2.5 J. Luterbacher and H. Zhang; Sect. 3: S. Coats, C. Colose, J. Jungclaus, W. Man, B. Otto-Bliesner, S. Stevenson, and D. Zanchettin; Sect. 4: B. Cook, A. Gallant, B. Konecky, A. Lopatka, D. Singh, N. Steiger and J. Tierney; Sect. 5: K. Anchukaitis, T. Ault, J. Mankin and S. Lewis. Lamont contribution #XXXX.



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



**Tables**

**Table 1:** PMIP3/CMIP5 and post-PMIP3/CMIP5 simulations of the CE and their associated forcings.

| **PMIP3/CMIP5 experiments** | | | |
|---|---|---|---|
| BCC-csm1-1* | (1x) 850-2005 | $SW^{15} \cdot V^{24} \cdot G^{30,33,34} \cdot A^{45} \cdot O^{60}$ | |
| CCSM4* | (1x) 850-2005 | $SW^{18} \cdot V^{24} \cdot G^{30,33,34} \cdot A^{45} \cdot L^{51} \cdot O^{60}$ | Landrum et al. (2013) |
| CSIRO-MK3L-1-2 | (1x) 851-2000 | $SW^{14} \cdot V^{25^*} \cdot G^{30,33,34} \cdot O^{60}$ | |
| FGOALS-s2* | (1x) 850-2005 | $SW^{18} \cdot V^{24} \cdot G^{30,33,34} \cdot A^{45} \, O^{60}$ | |
| GISS-E2-R | (8x) 850-2005 | $^*SW^{17} \cdot V^{25} \cdot G^{30,33,34} \cdot A^{45} \cdot L^{51} \cdot O^{60}$<br>$SW^{17} \cdot V^{24} \cdot G^{30,33,34} \cdot A^{45} \cdot L^{51} \cdot O^{60}$<br>$SW^{17} \cdot G^{30,33,34} \cdot A^{4} \cdot L^{51} \cdot O^{60}$<br>$SW^{18} \cdot V^{25} \cdot G^{30,33,34} \cdot A^{45} \cdot L^{51} \cdot O^{60}$<br>$SW^{18} \cdot V^{24} \cdot G^{30,33,34} \cdot A^{45} \cdot L^{52} \cdot O^{60}$<br>$SW^{18} \cdot G^{30,33,34} \cdot A^{4} \cdot L^{51} \cdot O^{60}$<br>$SW^{18} \cdot V^{25} \cdot G^{30,33,34} \cdot A^{45} \cdot L^{52} \cdot O^{60}$<br>$SW^{18} \cdot V^{24} \cdot G^{30,33,34} \cdot A^{45} \cdot L^{51} \cdot O^{60}$ | (4) |
| HadCM3* | (1x) 800-2000 | $SW^{14} \cdot V^{25} \cdot G^{30,32,34} \cdot A^{43} \cdot L^{51} \cdot O^{60}$ | Schurer et al. (2013) |
| IPSL-CM5A-LR* | (1x) 850-2005 | $SW^{15} \cdot V^{24} \cdot G^{30,33,34} \cdot O^{60}$ | |
| MIROC-ESM | (1x) 850-2005 | $SW^{16} \cdot V^{25} \cdot G^{30,34,39} \cdot O^{60}$ | (5) |
| MPI-ESM-P* | (1x) 850-2005 | $SW^{15} \cdot V^{25} \cdot G^{30,33,34} \cdot A^{45} \cdot L^{51} \cdot O^{60}$ | |
| **Post-PMIP3/CMIP5 experiments** | | | |
| CESM1(CAM5) | (13x) 850-2005*<br>(4x) 850-2005<br>(5x) 850-2005<br>(3x) 850-2005<br>(3x) 850-2005<br>(3x) 850-2005<br>(5x) 1850-2005 | $SW^{18} \cdot V^{24} \cdot G^{30,33,34} \cdot A^{45} \cdot L^{51} \cdot O^{60}$<br>$SW^{18}$<br>$V^{24}$<br>$G^{30,33,34}$<br>$L^{51}$<br>$O^{60}$<br>$A^{45}$ | Otto-Bliesner et al. (2016) |

(1)   Key for superscript indices in forcing acronyms:
      [ 1]   **Solar**:
      [14]   Steinhilber et al. (2009) spliced to Wang et al. (2005)
      [15]   Vieira and Solanki (2010) spliced to Wang et al. (2005)
[16]   Delaygue and Bard (2011) spliced to (Wang et al., 2005)
      [17]   Steinhilber et al. (2009)  spliced to Lean et al. (2005)
      [18]   Vieira et al. (2011) spliced to Lean et al. (2005)
      [ 2]   **Volcanic**:
      [24]   Gao et al. (2008). In the GISS-E2-R simulations this forcing was implemented twice as large as in Gao et al. (2008).
      [25]   Crowley and Unterman (2013). In the CSIRO simulation this forcing was implemented as a global-mean reduction in total solar
irradiance.



| | [ 3] | **GHGs**: |
|---|---|---|
| | [30] | Fluckiger et al. (1999); Fluckiger et al. (2002); Machida et al. (1995) |
| | [32] | Johns et al. (2003) |
| | [33] | Hansen and Sato (2004) |
| 2350 | [34] | MacFarling Meure et al. (2006) |
| | [39] | $CO_2$ diagnosed by the model. |
| | [4] | **Aerosols:** |
| | [43] | Johns et al. (2003) |
| | [45] | Lamarque et al. (2010) |
| 2355 | [5] | **Land use, land cover:** |
| | [51] | Pongratz et al. (2009) spliced to Hurtt et al. (2006) |
| | [52] | Kaplan (2011) |
| | [6] | **Orbital:** |
| | [60] | Berger (1978) |
| 2360 | | |





**Figures**

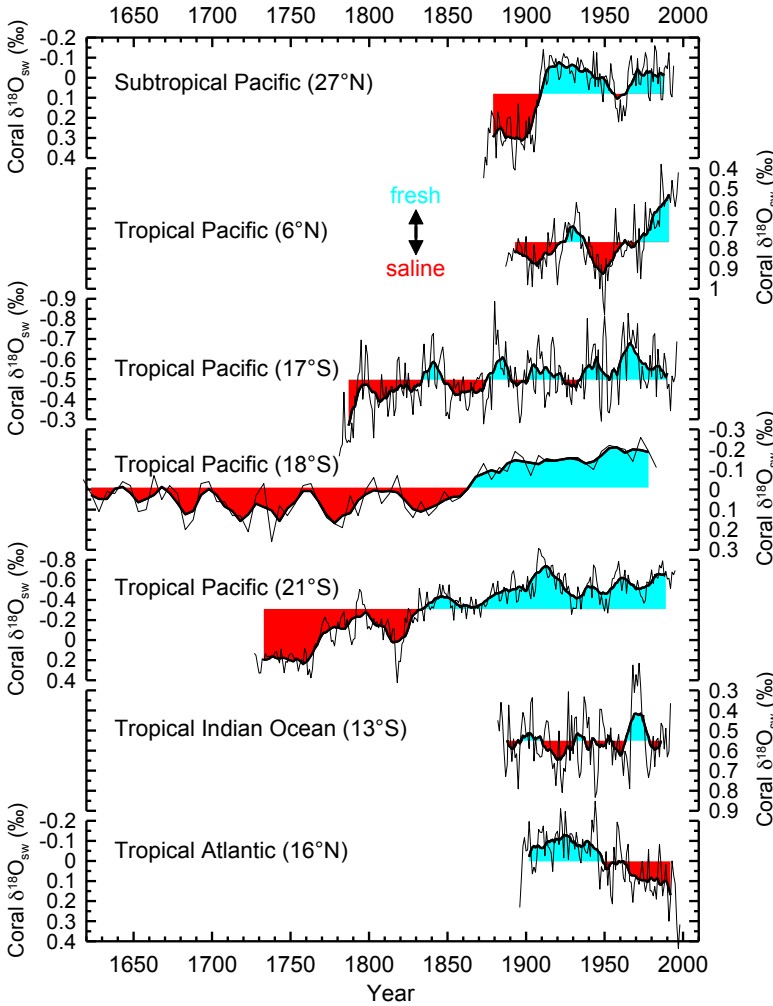

**Figure 1.** Annual reconstructions of seawater $\delta^{18}O$ ($\delta^{18}O_{sw}$) near the ocean surface, derived from subseasonal Sr/Ca and $\delta^{18}O$
measurements in shallow-water corals. Top to bottom: Ogasawara (Felis et al., 2009a, b) (average of Sr/Ca-$\delta^{18}O_{sw}$ and U/Ca-
$\delta^{18}O_{sw}$ reconstruction), Palmyra (Nurhati et al., 2011b; Nurhati et al., 2016), Fiji (Linsley et al., 2004; Linsley et al., 2006;
Linsley, 2017a), Great Barrier Reef (Hendy et al., 2002; Hendy et al., 2016) (pentannual resolution), Rarotonga (Linsley et al.,
2000; Ren et al., 2003; Linsley, 2017b), Mayotte (Zinke et al., 2008, 2016), and Guadeloupe (Hetzinger et al., 2010; Hetzinger
et al., 2017). Calculated $\delta^{18}O_{sw}$ is derived from original publications. Note that the absolute magnitude of changes in
reconstructed $\delta^{18}O_{sw}$ is dependent on the selected proxy-temperature relationships. Bold lines: 13-year running averages (Great
Barrier Reef: 3-point running average). The red-blue shading is centered on the mean of each record.







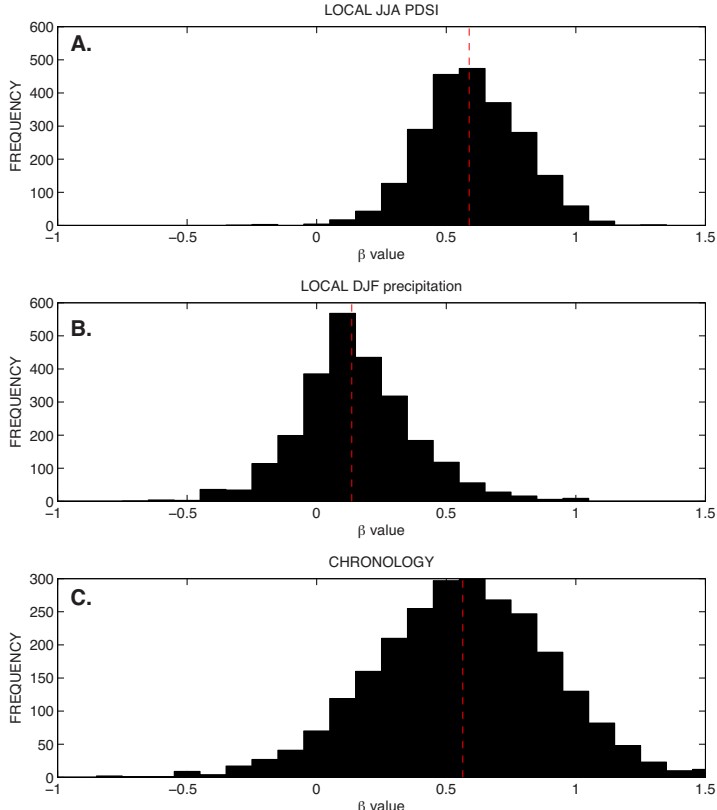

**Figure 2.** The slope of the multitaper spectrum expressed as β-value (Huybers and Curry, 2006) for climate and tree-ring
variables at locations with tree-ring chronologies from the North American Drought Atlas (Cook et al., 1999; Cook et al.,
2010b), the Monsoon Asia Drought Atlas (Cook et al., 2010a), the Old World Drought Atlas (Cook et al., 2015b), and the
Mediterranean and North Africa (Touchan et al., 2011; Touchan et al., 2014). (A) β-value frequency distribution for summer
(JJA) PDSI from (van der Schrier et al., 2011) at tree-ring chronology locations, (B) β-value frequency distribution for winter
(DJF) GPCC precipitation (Schneider et al., 2014) at tree-ring chronology locations, and (C) for the tree-ring chronologies
themselves. All data were normalized [0,1] over their common interval and their β-values were calculated using the multitaper
method (Thomson, 1982).



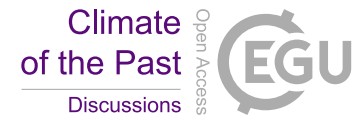

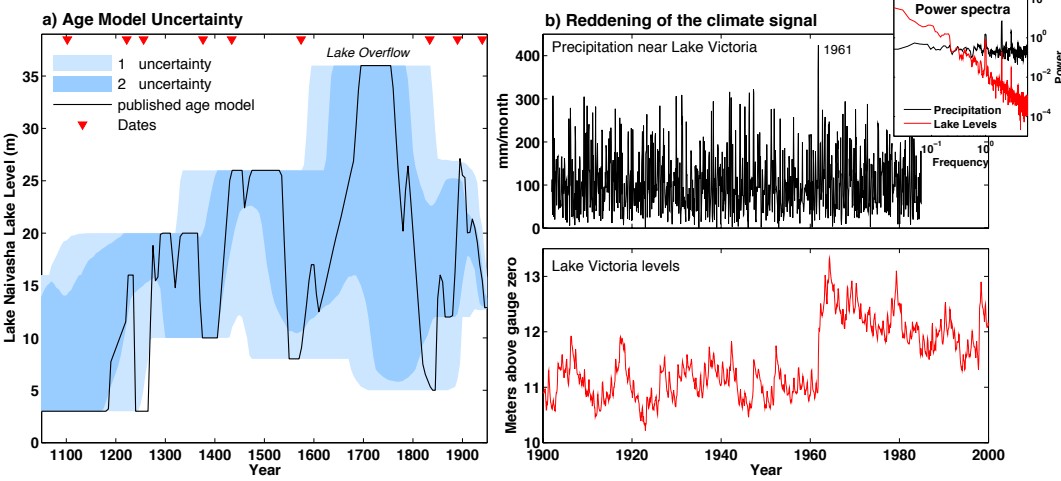

**Figure 3.** Examples of the effect of age-model uncertainty and spectral reddening on paleoclimate signatures in sedimentary archives. a) Age model uncertainty associated with the Lake Naivasha (East Africa) lake level record (Verschuren et al., 2000) spanning the last millennium. Data on the published age model are plotted in black. Blue regions represent the 1σ and 2σ uncertainty bounds for the dating of the data, applying a Monte Carlo method for iterating age model uncertainty (Anchukaitis and Tierney, 2013). Red triangles denote the locations of chronological constraints (dates). Note that the timing of lake overflow during the Little Ice Age may have occurred anytime between 1600-1800 CE. Figure after (Tierney et al., 2013). b) Comparison of historical monthly precipitation (from the Global Historical Climatology Network, Peterson and Vose (1997)) and lake level data (from Stager et al., 2007) for Lake Victoria, East Africa. Both the precipitation and the lake level data are from Jinja, Uganda. Note the step-change response of lake level to an extreme rainfall event in 1961. Inset shows a comparison of the power spectra; note the loss of high frequency and increase in low frequency power in the lake level data as compared to precipitation.



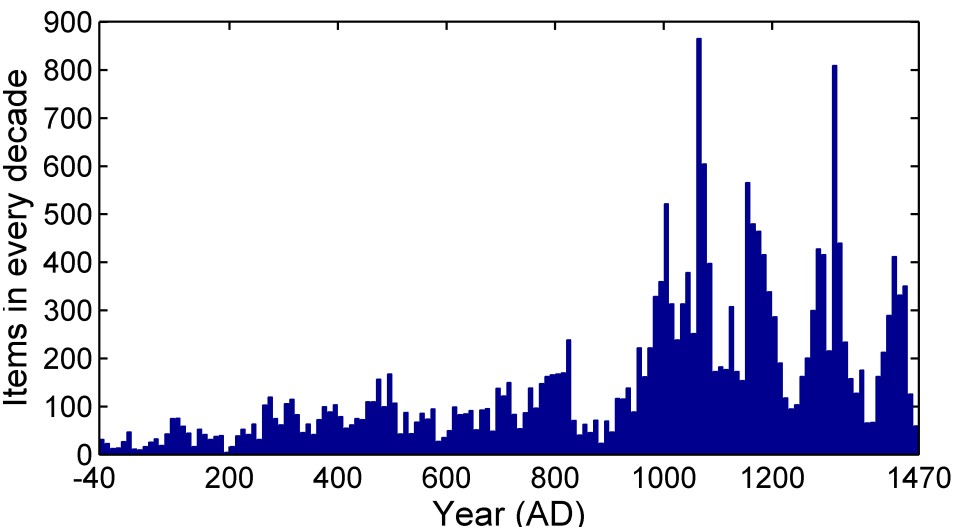


**Figure 4.** The amount of climatic information extracted from Chinese classical documents (a reproduction of Figure 7 in Ge et al. (2008b)).





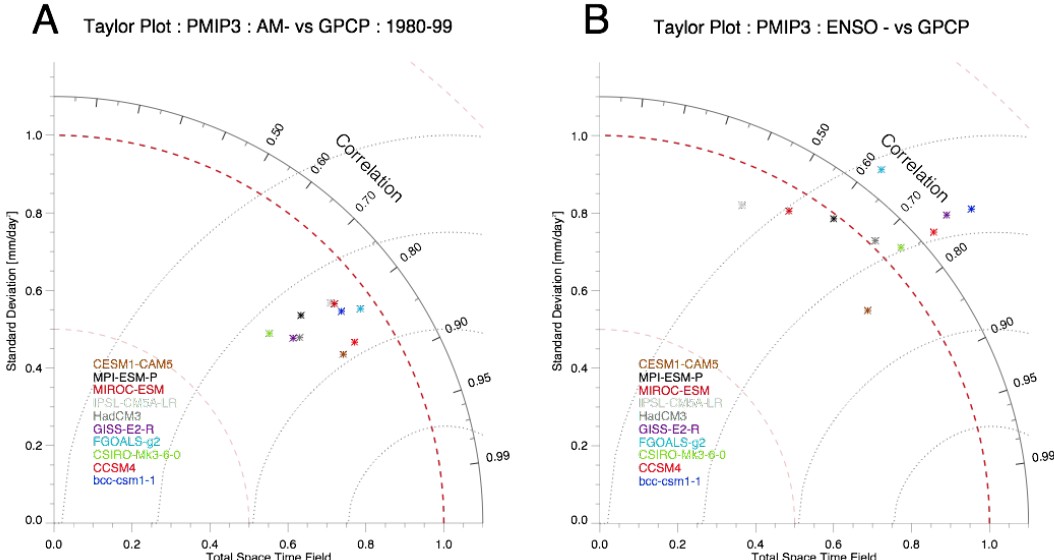

**Figure 5.** Taylor diagrams of global precipitation over land (Panel A) and ENSO (as defined by the Nino3.4 index) correlations to precipitation over land (Panel B). Each panel shows models that completed PMIP3 simulations of the last millennium as compared to precipitation observations from GPCP for the overlapping periods of 1980-1999 for Panel A and 1980-2014 for Panel B (not all models simulated the full period, all are at least 1980-2004).





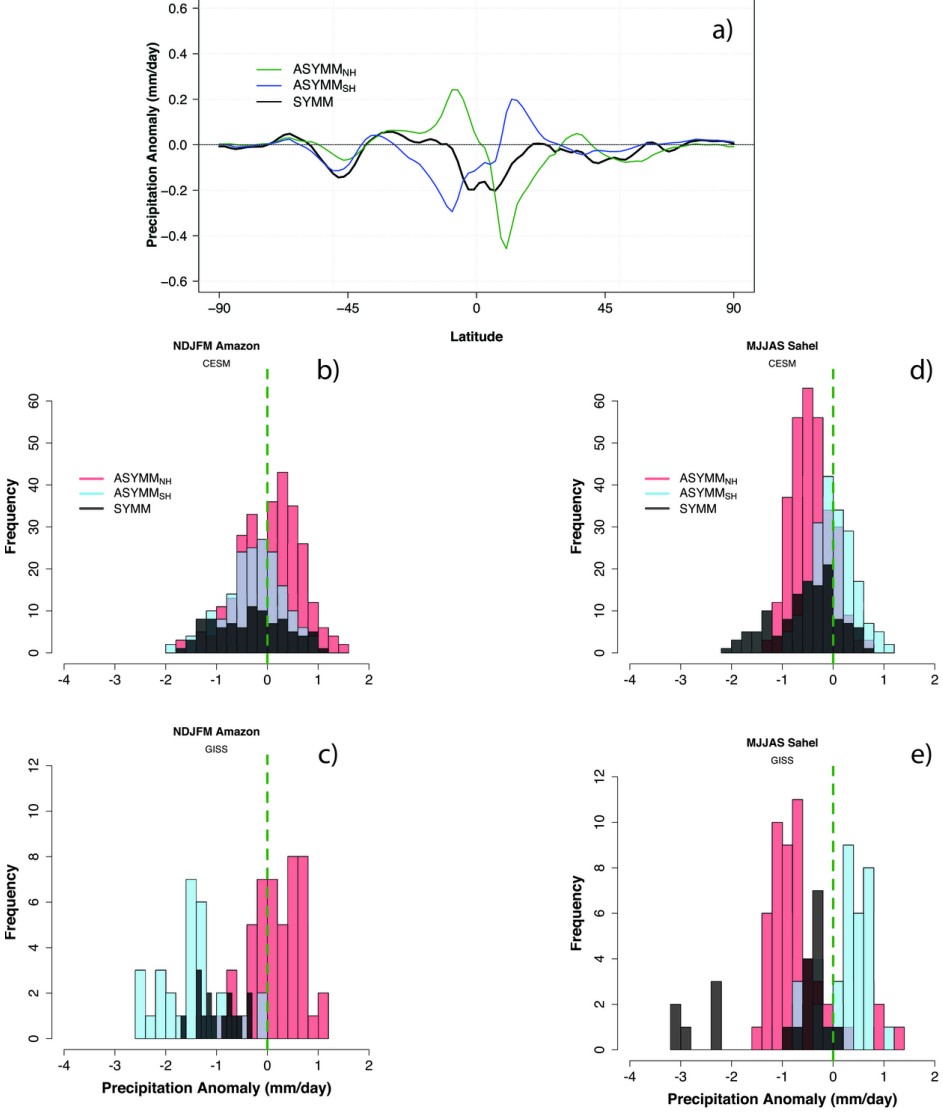

**Figure 6.** (**a**) Composite zonal-mean precipitation anomalies (mm/day) following volcanic eruptions with hemispherically symmetric loading (black) and for preferential aerosol loading in the NH (green) and SH (blue). The grouping of eruption categories follows the methodology in (Colose et al., 2016b) and results are shown for all-inclusive eruptions within 18 ensemble members in CESM LME. (**b**) Histogram showing frequency of precipitation response in CESM LME during the NDJFM season over the Amazon and (**d**) MJJAS in the Sahel. Results are for all NH eruptions (red), SH eruptions (aqua-blue) and symmetric events (black). (**c and e**) As in (b and d) except using NASA GISS ModelE2-R with 3 ensemble members, and larger forcing. Note that the forcing implementation in this version used the spatial structure in Gao et al. (2008) but with approximately two times the appropriate radiative forcing, as described in text.





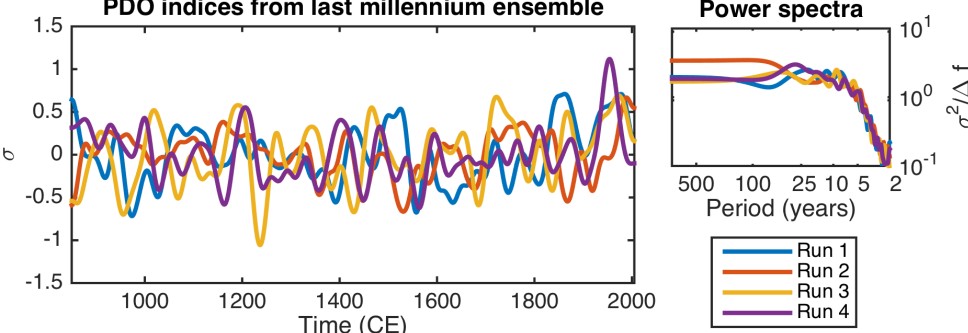


**Figure 7.** Time series (low-pass filtered to emphasize decadal variations) of the PDO from the four runs of the CESM LENS experiment (left), and power spectra of each run (right).




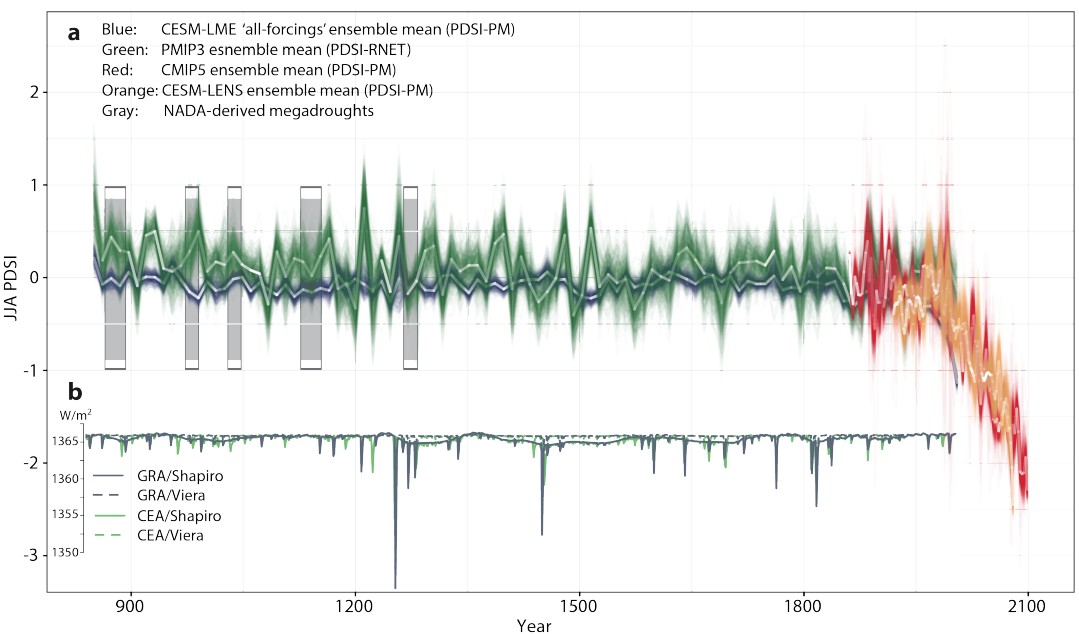

**Figure 8.** (a) JJA PDSI over the American Southwest for the CESM LME (blues), the CMIP5 historical and projection interval (red), the CESM LENS for the historical and projection interval (orange) and the PMIP3 net-radiation PDSI for the last-millennium and projection interval (green). For the CESM LME, the ensemble mean is shown for the all-forcings experiment.

For the PMIP3, CMIP5, and LENS, the ensemble mean is shown. For each of the ensemble means, a locally-weighted regression is bootstrapped 500 times to highlight the uncertainty in filtering. The median bootstrap value is indicated as a white line in each series, with the color opacity indicating the confidence of the true median value of the regression at that point. Grey bars indicate five largest megadroughts estimated by the NADA, all of which occurred during the Medieval Climate Anomaly. (b) The combinations of two volcanic (GRA (Gao et al., 2008) and CEA (Crowley and Unterman, 2013)) and two solar (Viera

(Vieira and Solanki, 2010) and Shapiro (Shapiro et al., 2011)) forcing data sets for the period 1000-2000 C.E. (from Coats et al., 2016a).



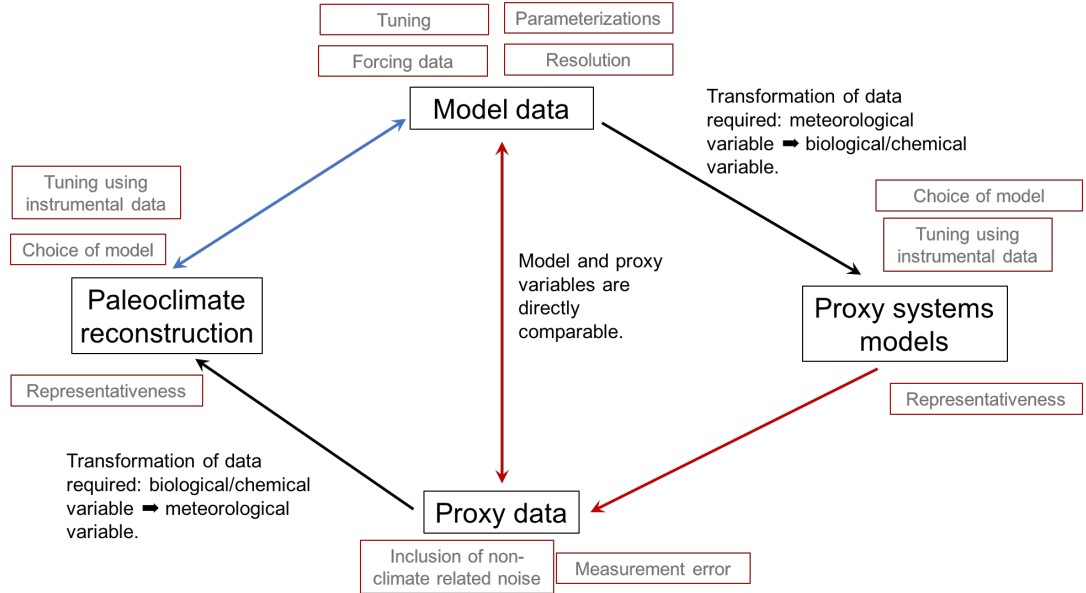


**Figure 9.** The black boxes show the data sources and the transformations between data sources that are required for paleoclimate proxy-model comparisons. The red boxes show the sources of uncertainty for each data source or data transformation step. The arrows show the directions in which any comparisons or data transformations take place. Following the data transformation or comparison, those sections with red arrows result in variables that are not explicitly associated with

hydroclimate and so further interpretation is required. Blue arrows show where data has been transformed into a variable that is directly relevant to hydroclimate.