# Peer review of "Comparing proxy and model estimates of hydroclimate variability and change over the Common Era"

_Climate of the Past, 2017_

## Referee Comment (RC1) · H. Xu (Referee) · 3 May 2017

**General Comments**

The authors reviewed the *state-of-the-art* hydroclimatic reconstructions, hydroclimatic modeling, and the comparison between the proxy-based reconstructions and model outputs of the last two millennia. They further reviewed the possibility to assess future hydroclimatic risks based on the past hydroclimatic reconstructions. In general, this paper merits in detailed and impersonal reviews/comments on both the advantages and uncertainties of proxy-based hydroclimatic reconstructions and model outputs, and it is clearly helpful to the readers to refresh their knowledge and understanding, and important to evaluate the fidelity of the climatic reconstruction, modeling outputs, and further prediction. I think a large number of audiences would be interested in the topics presented in this paper. Therefore I would be glad to suggest a publication of this paper on Climate of the Past.

The following specific points are for the authors' further consideration.

**Specific comments**

1. **Interpretation of the proxy indices**

The authors listed both the advantages and uncertainties of the proxy indices. This is very important! More examples may be clearer to the readers. For example, variations in $\delta^{18}O$ of corals are widely used to reconstruct sea surface temperature (SST). However, variable factors may influence the reliability of the $\delta^{18}O_{coral}$-based temperature reconstruction, such as the coral numbers, water depth, variable temperatures and sea water $\delta^{18}O$ values in different micro-environments, etc. One widely cited application is the reconstruction of SST based on $\delta^{18}O_{coral}$ over the central tropical Pacific Ocean (Cobb et al., 2003); it indicated a La Niña-like condition and an El Niño-like condition during the medieval period and little ice age (LIA), respectively, and such a SST pattern has been widely used in model simulations. However, increasing recent studies suggested a possibly inverse SST pattern to that inferred from $\delta^{18}O_{coral}$, i.e., an El Niño-like condition and a La Niña-like condition during the medieval period and LIA, respectively (see details in Xu et al., 2016). Providing the latter stands, a large number of model outputs perhaps need to be refreshed.

The recent years have witnessed an even more critical condition when interpreting the climatic significance of $\delta^{18}O$ in speleothem on decadal/multi-decadal to centennial timescales during the late Holocene. Sometimes, hydroclimatic changes inferred from different stalagmites even from a same cave are different. For example, the hydroclimatic conditions inferred from mud-layers in a stalagmite from KNI-5, northern Australia, are much different with those inferred from stalagmite $\delta^{18}O$ values from the same cave (see Fig. S9 in Denniston et al., 2015). Another example may be the differences between the late Holocene $\delta^{18}O$ series extracted from different stalagmites in Keshang Cave, northwestern China (Fig. 1). Variations in $\delta^{18}O$ were ascribed to changes in precipitation by Cheng et al. (2012). While Cai et al. (2017) gave more complicated interpretations; they ascribed the variations in $\delta^{18}O$ of the last ~2,000 years to temperature variations while those before were ascribed to changes in precipitation, leaving the readers a dilemma which one could be used to indicate the

regional climatic changes.

[Figure]

Fig.1. Comparison between late Holocene stalagmite $\delta^{18}O$ series from Keshang Cave, northwestern China (redrawn from Cheng et al., 2012 and Cai et al., 2017). The Green curve was developed by Cheng et al. (2012), and the pink, yellow, and blue curves were developed by Cai et al. (2017).

Closed-basin lake level changes are unique because they are first order recorders of past rainfall amounts (Verschuren et al., 2000; Xu et al., 2016; Goldsmith et al., 2017), and could provide a potential better way to reconstruct the past changes in precipitation after proper calibration of the evaporation. Similar methods can also be applied to reconstruct the past riverine runoff intensity, and to assess possible extreme flood risks based on the paleo- hydrological scenarios.

**2. Hydroclimatic contrasts between different regions**

Temperature variations are broadly synchronous between different regions due to the common large scale forcing, like solar forcing and volcanic aerosol forcing. However, unlike temperature, precipitations are influenced by variable factors, including atmospheric circulation, topography, underlying surface settings, etc. Hydroclimate, as a balance between precipitation and evaporation, could be more variable between different regions. Although orbital scale hydroclimatic trends have been reported to be similar between different regions, multi-decadal to centennial hydroclimatic changes are considerably different between different regions (e.g., Diaz et al., 2011; Graham et al., 2011; Xu et al., 2016). For example, increasing lines of evidence show late Holocene hydroclimatic contrasts between East Asian summer monsoon regions and Indian summer monsoon regions (see Xu et al., 2016 for details), and similar hydroclimatic contrasts have also been reported between the westerly-dominated Asian central arid zone and Asian monsoon areas (Chen et al., 2015). However, such hydroclimatic contrasts are not well reproduced by climatic modeling. More words about the hydroclimatic contrasts between different regions may help the readers, especially the policy-makers, to fully understand the features in hydroclimatic changes.

**3. Hydroclimatic forcings**

The influences of volcanic eruption on temperature were well reviewed in this paper. As the earth climates are influenced by variable factors, equal reviews of the role of solar forcing on temperatures and precipitations during the past ~2,000 years may further consummate this review paper.

**References mentioned in this comment**

1.  Cai, Y. J., Chiang, J. C. H., Breitenbach, S. F. M., et al., 2017. Holocene moisture changes in western China, Central Asia, inferred from stalagmites. Quaternary Science Reviews, 158, 15-28.

2.  Chen, J. H., Chen, F. H., Feng, S., et al., 2015. Hydroclimatic changes in China and surroundings during the Medieval Climate Anomaly and Little Ice Age: spatial patterns and possible mechanisms. Quaternary Science Reviews, 107, 98-111.

3.  Cheng, H., Zhang, P. Z., Spötl, C., et al., 2012. The climatic cyclicity in semiarid-arid central Asia over the past 500,000 years. Geophysical Research Letters, 39, doi:10.1029/2011GL050202.

4.  Cobb, K. M., Charles, C. D., Cheng. H., et al., 2003. El Niño/Southern Oscillation and tropical Pacific climate during the last millennium. Nature, 424, 271-276.

5.  Denniston, R. F., Villarini, G., Gonzales, A. N., et al., 2015. Extreme rainfall activity in the Australian tropics reflects changes in the El Niño/Southern Oscillation over the last two millennia. PNAS, 112(15), 4576-81. doi: 10.1073/pnas.1422270112.

6.  Diaz, H. F., Trigo, R., Hughes, M. K., et al., 2011. Spatial and temporal characteristics of climate in medieval times revisited. Bull. Am. Meteorol. Soc., 92, 1487-1500.

7.  Goldsmith, Y., Broecker, W. S., Xu, H., et al., 2017. Northward extent of East Asian monsoon covaries with intensity on orbital and millennial timescales. PNAS, 114 (8), 1817-1821.

8.  Graham, N. E., Ammann, C.M., Fleitmann, D., et al., 2011. Support for global climate reorganization during the "Medieval Climate Anomaly". Clim. Dynam., 37, 1217-1245.

9.  Verschuren, D., Laird, K. R., Cumming, B. F., 2000. Rainfall and drought in equatorial east Africa during the past 1,100 years. Nature, 403, 410-414.

10. Xu, H., Lan, J., Sheng, E., et al., 2016. Hydroclimatic contrasts over Asian monsoon areas and linkages to tropical Pacific SSTs. Scientific Reports, 6, 33177; doi: 10.1038/srep33177.

---

## Short Comment (SC1) · 3 Jun 2017

The PAGES Data Stewardship Integrative Activity seeks to advance best practices for sharing data generated and assembled as part of all PAGES-related activities. As part of this activity, a team of reviewers has been constituted for the "Climate of the Past 2000 years" Special Issue. The data team is reviewing the data handling within each of the CP-Discussion papers in relation to the CP data policy and current best practices. The team has identified essential and recommended additions for each paper, with the goal of achieving a high and consistent level of data stewardship across the 2k Special Issue. We recognize that an additional effort will likely be required to

meet the high level of data stewardship envisaged, and we appreciate the dedication and contribution of the authors. This includes the use of Data Citations (see example in supplement). We ask authors to respond to our comments as part of the regular open interactive discussion. If you have any questions about PAGES Data Stewardship principles, please contact any of us directly.

Best wishes for the success of your paper,

2k Special Issue Data Review Team (Darrell Kaufman, Nerilie Abram, Belen Martrat, Raphael Neukom, Scott St. George) and ex-officio team members (Marie-France Loutre, Lucien von Gunten)

For this paper: The "Data Availability" section is excellent, with all essential elements included. We commend the authors for their diligent data stewardship.

We suggest that you also submit to a public repository the data used to create the: (a) histograms in Fig 2, (b) Taylor diagrams in Fig 5, (c) histograms in Fig 6, and (d) JJA PDSI time series in Fig 8a. Add the Data Citation (or URL link) to these datasets to the "Data Availability" section of the paper.

Please also note the supplement to this comment:
http://www.clim-past-discuss.net/cp-2017-37/cp-2017-37-SC1-supplement.pdf

---

## Referee Comment (RC2) · Anonymous Referee #2 · 6 Jun 2017

The manuscript presents a nice overview of the present knowledge of the hydroclimate variability over the past millennium from the joint perspective of climate simulations and proxy reconstructions. It illustrates some of the problems that arise when comparing both sources of information about past hydroclimate variability. I see very clear positive aspects in this manuscript , and I certainly recommend it for publication in Climate of the Past. I just have some suggestions that the authors may want to consider in a revised version, but most of them are more a matter of perspective. I also want to congratulate the leading authors for accomplishing a rather difficult task, namely to web a coherent and actually very informative text out of a workshop with so many participants and probably very diverse interests.

My suggestions follow:

1. When listing the most important limitations and deficiencies that climate models still show and that can be important for the simulation of hydroclimate, I missed that extratropical blocking is not part of this list (around line 540) . It is rather well known that climate models still suffer from clear limitations in this regard, although they are getting better at that. Nevertheless, blocking is a very important phenomenon for seasonal drought in extratropical land masses. North Atlantic blocking is not particularly well represented in virtually all CMIP5 models and it is very relevant for drought episodes in Europe.

2. Around line 590 the manuscript discusses the mechanism 'rich gets richer' and essentially accepts it as established truth. However, although this mechanism has been derived from basic theoretical considerations for the tropical oceans (Held and Soden, 2006), it is still far from clear that it is the main mechanism that can explain the response of hydroclimate to external forcing, here and in other areas. For instance, other later studies have looked into CMIP3/5 simulations and found that this mechanism may be overlaid by others that also impact on the patterns of change of hydroclimate, so that in the end the correlation between the pattern of response to CO2 and the climatological pattern of hydroclimate is almost zero, even in the tropics. In the extratropics, the situations may be even more complex, with multiple factors (or different aspects of the same factor) interacting in clear ways, ranging from increased humidity, shifts in the strom tracks, expansion of the Hadley cells, and shifts in SSTs gradients. Perhaps the authors may want to briefly discuss the papers mentioned below and others. I am aware that the space is limited and this is a very complex question, but I think it would be important to at least convey the message that the 'rich-gets-richer mechanism, though plausible and brilliant when it was put forward, may not be the whole truth.

Chadwick et al, Spatial Patterns of Precipitation Change in CMIP5: Why the Rich Do Not Get Richer in the Tropics doi:10.1175/JCLI-D-12-00543.1

Chou et al.Evaluating the "Rich-Get-Richer" Mechanism in Tropical Precipitation Change under Global Warming. doi:10.1175/2008JCLI2471.1

Huang et al: Patterns of the seasonal response of tropical rainfall to global warming, doi: DOI: 10.1038/NGEO1792

Sheff and Frierson, Robust future precipitation declines in CMIP5the poleward expansion of model subtropical largely reflect dry zones. doi:10.1029/2012GL052910

3. Around line 593

' In the tropics, areas experiencing post-eruption drying coincide with climatologically wet regions, while dry regions get wetter on average, but the changes are spatially heterogeneous. This pattern is of opposite sign to, but physically consistent with, projections under global warming.'

This sentence sounds a bit strange at first sight (consistent and yet of opposite sign). Perhaps reformulate as 'consistent with projections under increased greenhouse gas forcing, since volcanic forcing has the opposite sign'

4. I was also a bit surprised that regional modelling was very briefly discussed (just a couple of sentences), whereas other issues that are also not very well developed are considered in much more detailed was , for instance estimation of future drought risks. This may reflect missing regional climate modellers among the authors, but the manuscript gives the impression that regional climate modelling is considered not very important for simulations of hydroclimate variability or even for the comparison of proxies. Given the stress on the limited resolution of global models, this seems odd.

5. A few thoughts on data assimilation: data assimilation is indeed very important for weather prediction and in general for any kind of prediction. For this purpose, what counts is a skillful prediction and the understanding of the physical mechanisms remains a bit in the background. Thus it is permissible to violate the model physics with data assimilation if this really leads to an improvement (Data assimilation is also important to initialise the model, but I think this is not the issue here). My point is that, in the context of paleoclimate, the benefits of data assimilation are much less clear than in weather prediction. It may serve, for instance, to identify physical inconsistencies in the proxies (when the model 'rejects' to accept information from two diverging proxies, or it may help to physically explain a particular proxy configuration if DA is able to bring the model to a consistent state that is compatible with all proxy records. However, it is not that clear to me that DA can help ' assess the physical causes of past climate'. Actually, DA produces an output that deviates from what the model wants to provide, or when DA assimilation is used to drive the model towards the observed trajectory, it also bends, nudges, or combines model physics with observations. The results is certainly not as physically consistent as the raw model output, and it is unclear how it can help to understand mechanisms. It does help for predictions ( or in this case reconstructions) though

6 . Line 795 ' warm climate intervals, and the mean state of the Indo-Pacific system. The paleoclimate record showed that the 20th century was actually dominated by a strong drying trend unprecedented in the last millennium, indicating that greenhouse gases may in fact be driving the East African region toward a drier state; the wetter prediction of the models associated with greenhouse gas forcing therefore deserves further investigation'

The paragraph sounds a bit strange. Why use the paleoclimate record in the 20th century ? The conclusion of this paragraph also sounds a bit weak (deserves further investigation), and this gives me the opportunity to comment on an aspect that seems to be missing in the whole manuscript. Are paleo reconstructions useful to identify 'bad models' that should not be used for climate projections ? Is this not what reducing projection uncertainty means in the end ?

7. Line 1080. It is nice to have the conclusions summarized in a few sentences, but the first conclusion looks unclear or weak: ' Expectations of temporal or spatial consistency between proxies and models should be critically evaluated' . I would suggest to clearly

state that temporal consistency can only be expected if and only if the variability is externally forced

8. Colose, C. M., LeGrande, A. N., and Vuille, M.: The influence of volcanic eruptions on the climate of tropical South America during the last millennium in an isotope-enabled general circulation model, Climate of the Past, 12, 961-979, 10.5194/op-12-961-2016, 2016a.

There is a typo in the doi of this paper. It should be cp and not op.

---

## Author Comment (AC1) · 17 Jul 2017

The comment was uploaded in the form of a supplement:
https://www.clim-past-discuss.net/cp-2017-37/cp-2017-37-AC1-supplement.pdf

---

## Author Response (AR1)

We thank the reviewers for their time reviewing our manuscript and their insightful comments and suggestions.

Below we provide the original responses to each of the reviewers' comments that indicated our plans to revise the manuscript and subsequently the specific changes that we have made to the revised manuscript. Reviewer comments are shown in *italic*. Our original responses are shown in **bold** and the updated descriptions of how we have revised the manuscript are shown in **bold blue text**. Quotations from the original manuscript are shown in ***bold italics*** and quotations from the revised manuscript are shown in ***bold blue italics***. Line numbers provided in black are from the original manuscript and line numbers in blue are for the revised manuscript.

**Reviewer 1**

*General Comments*
*The authors reviewed the state-of-the-art hydroclimatic reconstructions, hydroclimatic modeling, and the comparison between the proxy-based reconstructions and model outputs of the last two millennia. They further reviewed the possibility to assess future hydroclimatic risks based on the past hydroclimatic reconstructions. In general, this paper merits in detailed and impersonal reviews/comments on both the advantages and uncertainties of proxy-based hydroclimatic reconstructions and model outputs, and it is clearly helpful to the readers to refresh their knowledge and understanding, and important to evaluate the fidelity of the climatic reconstruction, modeling outputs, and further prediction. I think a large number of audiences would be interested in the topics presented in this paper. Therefore I would be glad to suggest a publication of this paper on Climate of the Past.*

*The following specific points are for the authors' further consideration.*

*Specific comments*
1. *Interpretation of the proxy indices*
   *The authors listed both the advantages and uncertainties of the proxy indices. This is very important! More examples may be clearer to the readers. For example, variations in δ18O of corals are widely used to reconstruct sea surface temperature (SST). However, variable factors may influence the reliability of the δ18O coral-based temperature reconstruction, such as the coral numbers, water depth, variable temperatures and sea water δ18O values in different micro-environments, etc. One widely cited application is the reconstruction of SST based on δ18O coral over the central tropical Pacific Ocean (Cobb et al., 2003); it indicated a La Niña-like condition and an El Niño-like condition during the medieval period and little ice age (LIA), respectively, and such a SST pattern has been widely used in model simulations. However, increasing recent studies suggested a possibly inverse SST pattern to that inferred from δ18O coral, i.e., an El Niño-like condition and a La Niña-like condition during the medieval period and LIA, respectively (see details in Xu et al., 2016). Providing the latter stands, a large number of model outputs perhaps need to be refreshed.*

*The recent years have witnessed an even more critical condition when interpreting the climatic significance of δ18O in speleothem on decadal/multi-decadal to centennial timescales during the late Holocene. Sometimes, hydroclimatic changes inferred from different stalagmites even from a same cave are different. For example, the hydroclimatic conditions inferred from mud-layers in a stalagmite from KNI-5, northern Australia, are much different with those inferred from stalagmite δ18O values from the same cave (see Fig. S9 in Denniston et al., 2015). Another example may be the differences between the late Holocene δ18O series extracted from different stalagmites in Keshang Cave, northwestern China (Fig. 1). Variations in δ18O were ascribed to changes in precipitation by Cheng et al. (2012). While Cai et al. (2017) gave more complicated interpretations; they ascribed the variations in δ18O of the last ~2,000 years to temperature variations while those before were ascribed to changes in precipitation, leaving the readers a dilemma which one could be used to indicate the regional climatic changes.*

*Closed-basin lake level changes are unique because they are first order recorders of past rainfall amounts (Verschuren et al., 2000; Xu et al., 2016; Goldsmith et al., 2017), and could provide a potential better way to reconstruct the past changes in precipitation after proper calibration of the evaporation. Similar methods can also be applied to reconstruct the past riverine runoff intensity, and to assess possible extreme flood risks based on the paleo-hydrological scenarios.*

**While the reviewer does not explicitly suggest changes to the manuscript, we will work to include the listed citations (and other relevant citations) and reflect some of this discussion in the revised version of the manuscript.**

**In response to the reviewer's comments in the first paragraph above, we now include the following sentence in the revised manuscript (Lines 168-169):**

**Interpretations of coral records therefore can be complicated by sampling factors such as water depth and the temperature and $\delta^{18}O_{sw}$ of specific microenvironments.**

**We now include citations for the noted references in the second paragraph at Line 370 in the revised manuscript and Xu et al. (2016) is now cited at Line 390 in response to the comments in the last paragraph (the other noted references were already cited there).**

2. *Hydroclimatic contrasts between different regions*
*Temperature variations are broadly synchronous between different regions due to the common large-scale forcing, like solar forcing and volcanic aerosol forcing. However, unlike temperature, precipitations are influenced by variable factors, including atmospheric circulation, topography, underlying surface settings, etc. Hydroclimate, as a balance between precipitation and evaporation, could be more variable between different regions. Although orbital scale hydroclimatic trends have been reported to be similar between different regions, multi-decadal to centennial hydroclimatic changes are considerably different between different regions (e.g., Diaz et al., 2011; Graham et al.,*

*2011; Xu et al., 2016). For example, increasing lines of evidence show late Holocene hydroclimatic contrasts between East Asian summer monsoon regions and Indian summer monsoon regions (see Xu et al., 2016 for details), and similar hydroclimatic contrasts have also been reported between the westerly-dominated Asian central arid zone and Asian monsoon areas (Chen et al., 2015). However, such hydroclimatic contrasts are not well reproduced by climatic modeling. More words about the hydroclimatic contrasts between different regions may help the readers, especially the policy-makers, to fully understand the features in hydroclimatic changes.*

**Once again, there are no explicitly directed changes in this comment, but we will work to include the listed references and mention how regional contrasts in hydroclimate are relevant within our discussion.**

**The revised manuscript now reads as follows in the Introduction and includes the noted references (Lines 132-137):**

*Many of the same technical details associated with proxy-model comparisons of temperature (PAGES 2k-PMIP3 Group et al., 2015) are also applicable to hydroclimate. The latter nevertheless requires unique considerations before meaningful comparisons can be made, in part because of the many different processes and variables that are related to hydroclimate and because the spatial coherence of hydroclimate is more limited than temperature (e.g. Diaz et al., 2011; Graham et al., 2011; Chen et al., 2015; Swann et al., 2016; Xu et al., 2016; Mankin et al., 2017). Addressing these and other issues is an aim of contemporary research.*

3. *Hydroclimatic forcings*
   *The influences of volcanic eruption on temperature were well reviewed in this paper. As the earth climates are influenced by variable factors, equal reviews of the role of solar forcing on temperatures and precipitations during the past ~2,000 years may further consummate this review paper.*

**We give the following motivation for our focus on volcanic forcings in the original manuscript (Lines 583-588):**

**"Volcanic eruptions are the most important external radiative forcing at the global scale during the pre-industrial interval of the CE. Although the impacts persist only for a few years, volcanic forcing from multiple events constitutes the most important contributor to the global centennial-scale cooling observed from 850-1800 C.E., based on comparisons of single forcing experiments (Otto-Bliesner et al., 2016) or through the use of all-forcing simulations in which the contributions from individual forcings are isolated offline (Atwood et al., 2016). For this reason, volcanic eruptions are particularly important for gauging the forced hydroclimatic response in last-millennium simulations."**

**It is true that the lack of a strong response to solar forcing over the last millennium in climate model simulations does not necessarily mean that solar influence on the actual climate is similarly weak; the models could of course be wrong, or, as we discuss in the**

manuscript (Lines 718-721), the reconstructions of solar variability could be flawed. Given the current state of understanding, however, we believe that our focus on volcanic forcing is warranted. We nevertheless will include a brief discussion on solar forcing of hydroclimate during the Common Era in the revised manuscript within Section 3.

We have made multiple changes to the revised manuscript in response to this comment. The title of subsection 3.2 is now "Natural Forcing of Hydroclimate" instead of the version in the original manuscript that only referred to volcanic forcing. The introductory lines in the revised version of subsection 3.2 now note both volcanic and solar forcing as aspects of consideration (Lines 598-599-611) and an additional paragraph on solar forcing is now included at the end of the section (Lines 679-695).

**Reviewer 2**

*For this paper: The "Data Availability" section is excellent, with all essential elements included. We commend the authors for their diligent data stewardship. We suggest that you also submit to a public repository the data used to create the: (a) histograms in Fig 2, (b) Taylor diagrams in Fig 5, (c) histograms in Fig 6, and (d) JJA PDSI time series in Fig 8a. Add the Data Citation (or URL link) to these datasets to the "Data Availability" section of the paper.*

We are unclear on this specific directive because we have cited the public data sources for the tree-ring data in Figure 2 (ITRDB and NCEI) and the model data in Figures 5, 6 and 8 (ESGF and NCAR). If the reviewer is suggesting that we include the various data plotted in the figures (e.g. the beta values in Figure 2 or the correlations in Figure 5), we will make those additional data files available. Regarding the JJA PDSI data in Figure 8a, we will provide all of the mean PDSI time series for the American Southwest that are represented in the figure on a supplementary website constructed for the paper. Pending clarification from the editor, we will include the beta values and other small data files used to construct Figures 2, 5 and 6 on the same website.

We now note the following in the Data Availability section of the revised manuscript (Lines 1196-1199):

> *Should this manuscript be accepted for publication, we will make the processed data used to construct Figures 2, 5, 6 and 8 publicly available through the research-sharing site Zenodo, which specifically provides doi's for archived data. The data links established through Zenodo will be posted on the workshop website that was constructed through Github (http://pages2kpmip3.github.io/).*

**Reviewer 3**

*The manuscript presents a nice overview of the present knowledge of the hydroclimate variability over the past millennium from the joint perspective of climate simulations and proxy reconstructions. It illustrates some of the problems that arise when comparing both sources of information about past hydroclimate variability. I see very clear positive aspects in this manuscript , and I certainly recommend it for publication in Climate of the Past. I just have some suggestions that the authors may want to consider in a revised version, but most of them are more a matter of perspective. I also want to congratulate the leading authors for*

*accomplishing a rather difficult task, namely to web a coherent and actually very informative text out of a workshop with so many participants and probably very diverse interests.*

*My suggestions follow:*

*1. When listing the most important limitations and deficiencies that climate models still show and that can be important for the simulation of hydroclimate, I missed that extratropical blocking is not part of this list (around line 540). It is rather well known that climate models still suffer from clear limitations in this regard, although they are getting better at that. Nevertheless, blocking is a very important phenomenon for seasonal drought in extratropical land masses. North Atlantic blocking is not particularly well represented in virtually all CMIP5 models and it is very relevant for drought episodes in Europe.*

**We agree that extratropical blocking should be noted as a model limitation. This additional consideration will be noted in the revised manuscript.**

**The revised manuscript now lists atmospheric blocking as a phenomenon that is not well represented in climate models (Lines 557-558).**

*2. Around line 590 the manuscript discusses the mechanism 'rich gets richer' and essentially accepts it as established truth. However, although this mechanism has been derived from basic theoretical considerations for the tropical oceans (Held and Soden, 2006), it is still far from clear that it is the main mechanism that can explain the response of hydroclimate to external forcing, here and in other areas. For instance, other later studies have looked into CMIP3/5 simulations and found that this mechanism may be overlaid by others that also impact on the patterns of change of hydroclimate, so that in the end the correlation between the pattern of response to CO2 and the climatological pattern of hydroclimate is almost zero, even in the tropics. In the extratropics, the situations may be even more complex, with multiple factors (or different aspects of the same factor) interacting in clear ways, ranging from increased humidity, shifts in the strom tracks, expansion of the Hadley cells, and shifts in SSTs gradients. Perhaps the authors may want to briefly discuss the papers mentioned below and others. I am aware that the space is limited and this is a very complex question, but I think it would be important to at least convey the message that the 'rich-gets-richer mechanism, though plausible and brilliant when it was put forward, may not be the whole truth.*

*Chadwick et al, Spatial Patterns of Precipitation Change in CMIP5: Why the Rich Do Not Get Richer in the Tropics doi:10.1175/JCLI-D-12-00543.1*

*Chou et al.Evaluating the "Rich-Get-Richer" Mechanism in Tropical Precipitation Change under Global Warming. doi:10.1175/2008JCLI2471.1*

*Huang et al: Patterns of the seasonal response of tropical rainfall to global warming, doi: DOI: 10.1038/NGEO1792*

*Sheff and Frierson, Robust future precipitation declines in CMIP5the poleward expansion of model subtropical largely reflect dry zones. doi:10.1029/2012GL052910 3.*

**We of course agree with the reviewer's point and will revise to highlight more recent work identifying where the Held and Soden paradigm is insufficient (this is particularly the case over land). We will highlight this point, and the listed papers in the revised version of the manuscript.**

**We now include the following sentence in the revised manuscript (Lines 617-620):**

*It also must be noted that the general pattern of wet regions getting wetter and dry regions getting drier in response to warming forced by increasing greenhouse gases is not universally applicable and can break down particularly over land and in the tropics (Chou et al., 2009; Scheff and Frierson, 2012; Chadwick et al., 2013; Huang et al., 2013; Byrne and O'Gorman, 2015).*

*3. Around line 593*
        *'In the tropics, areas experiencing post-eruption drying coincide with climatologically wet regions, while dry regions get wetter on average, but the changes are spatially heterogeneous. This pattern is of opposite sign to, but physically consistent with, projections under global warming.'*

*This sentence sounds a bit strange at first sight (consistent and yet of opposite sign). Perhaps reformulate as 'consistent with projections under increased greenhouse gas forcing, since volcanic forcing has the opposite sign'*

**This sentence will be revised as suggested by the reviewer.**

**This sentence now reads as follows in the revised manuscript (Lines 613-617):**

*In the tropics, areas experiencing post-eruption drying coincide with climatologically wet regions, while dry regions get wetter on average, but the changes are spatially heterogeneous; a similar pattern also has been noted over Europe (Rao et al., 2017). These responses are physically consistent with future global warming projections, but of opposite sign because volcanoes and greenhouse gases have contrasting influences on radiative forcing.*

*4. I was also a bit surprised that regional modelling was very briefly discussed (just a couple of sentences), whereas other issues that are also not very well developed are considered in much more detailed was, for instance estimation of future drought risks. This may reflect missing regional climate modellers among the authors, but the manuscript gives the impression that regional climate modelling is considered not very important for simulations of hydroclimate variability or even for the comparison of proxies. Given the stress on the limited resolution of global models, this seems odd.*

**This was indeed an oversight. While there are fewer comparisons between regional model simulations and proxies, there is literature relevant to the subject that should be**

**highlighted in our review. We will include a discussion of work on such comparisons in the revised version of the manuscript.**

**Regional models are now further discussed in the revised manuscript at Lines 561-566 and Lines 748-753.**

*5. A few thoughts on data assimilation: data assimilation is indeed very important for weather prediction and in general for any kind of prediction. For this purpose, what counts is a skillful prediction and the understanding of the physical mechanisms remains a bit in the background. Thus it is permissible to violate the model physics with data assimilation if this really leads to an improvement (Data assimilation is also important to initialise the model, but I think this is not the issue here). My point is that, in the context of paleoclimate, the benefits of data assimilation are much less clear than in weather prediction. It may serve, for instance, to identify physical inconsistencies in the proxies (when the model 'rejects' to accept information from two diverging proxies, or it may help to physically explain a particular proxy configuration if DA is able to bring the model to a consistent state that is compatible with all proxy records. However, it is not that clear to me that DA can help 'assess the physical causes of past climate'. Actually, DA produces an output that deviates from what the model wants to provide, or when DA assimilation is used to drive the model towards the observed trajectory, it also bends, nudges, or combines model physics with observations. The results is certainly not as physically consistent as the raw model output, and it is unclear how it can help to understand mechanisms. It does help for predictions (or in this case reconstructions) though.*

**While the reviewer does not require any specific revisions based on the points that are raised, we wish to address some of them. It is perhaps most important to point out that the goals of DA for paleoclimate are different than those associated with weather prediction. Instead of providing a prediction, DA provides a statistical framework for constraining reconstructions of the past using climate models and for estimating multiple climate variables in the reconstructions. DA therefore provides insight into the physical causes of past climate by optimally merging information from both proxies and climate models to simultaneously reconstruct multiple dynamical variables. We will revise the manuscript to better clarify these points.**

**We have further clarified the motivation for paleoclimate DA in the revised manuscript as follows (Lines 989-995):**

> *Data assimilation (DA) is an emerging, promising and specific means of proxy-model comparison, which, among multiple benefits, can incorporate the use of water isotope modeling and PSMs. While paleoclimate DA is based on the same methodological framework as other DA applications, its motivations are typically different. DA for paleoclimate provides a coherent statistical framework for optimally fusing proxy information with the dynamical constraints of climate models, for providing many climate variables in the reconstruction, for coherently modeling proxy systems, and for explicitly accounting for some of the sources of uncertainty inherent in reconstructions (Goosse et al., 2010; Luterbacher et al., 2010; Widmann et al., 2010; Goosse et al., 2012b; Steiger et al., 2014; Hakim et al., 2016; Matsikaris et al., 2016).*

*6. Line 795: 'warm climate intervals, and the mean state of the Indo-Pacific system. The paleoclimate record showed that the 20th century was actually dominated by a strong drying trend unprecedented in the last millennium, indicating that greenhouse gases may in fact be driving the East African region toward a drier state; the wetter prediction of the models associated with greenhouse gas forcing therefore deserves further investigation'*

*The paragraph sounds a bit strange. Why use the paleoclimate record in the 20th century? The conclusion of this paragraph also sounds a bit weak (deserves further investigation), and this gives me the opportunity to comment on an aspect that seems to be missing in the whole manuscript. Are paleo reconstructions useful to identify 'bad models' that should not be used for climate projections? Is this not what reducing projection uncertainty means in the end?*

**We will revise this paragraph to make our arguments more clear. Regarding constraints on future projections, we do discuss the topic in the manuscript within the Conclusions and in Section 5 when we discuss rescaling model output based on proxy comparisons. Constraining future projections are indeed an important goal of proxy-model comparisons and we will revise the manuscript to make this clearer. The framework for such constraints nevertheless is still an area of active research. We also note that the presence of incomplete information in the error and bias structures of climate models, and the records against which we compare them, make it difficult to justify moving beyond a "one model, one vote" approach (e.g. Weigel et al., *J. Clim.*, 2010; Knutti et al., *J. Clim.*, 2010). Observations are nevertheless beginning to be used in this context (e.g. Knutti et al., *GRL*, 2017), suggesting that the use of paleoclimate records to constrain future projections, though saddled by more uncertainty, may be possible in the near future.**

**The noted sentence has been changed in the revised manuscript as follows (Lines 860-863):**

> ***The paleoclimate record showed that the strong 20th century drying trend was unprecedented over the last millennium, indicating that greenhouse gases may in fact be driving the East African region toward a drier state; model predictions of wetter conditions in the 21st century due to greenhouse gas forcing therefore stand in contrast to this finding and require further vetting.***

**To address the reviewer's larger point, we also conclude the manuscript with the following text (Lines 1169-1174):**

> ***The framework for such constraints is indeed an area of active research. The presence of incomplete information in the error and bias structures of climate models, and the proxy records against which we compare them, do make it difficult to move beyond a traditional equal-weighting scheme in the representation of model ensembles (Knutti et al., 2009; Weigel et al., 2010). Historical observations are nevertheless beginning to be used in this context (Knutti et al., 2017) and therefore suggest that the use of paleoclimate records to constrain future projections may indeed be within reach.***

*7. Line 1080. It is nice to have the conclusions summarized in a few sentences, but the first conclusion looks unclear or weak: 'Expectations of temporal or spatial consistency between proxies and models should be critically evaluated.' I would suggest to clearly state that temporal consistency can only be expected if and only if the variability is externally forced.*

**We will clarify this conclusion in keeping with the reviewer's suggestion.**

**The conclusion has been modified as follows in the revised manuscript (Lines 1155-1156):**

[revised manuscript text omitted]

---

## Author Response (AR2)

**Second Response File – Climate of the Past Manuscript cp-2017-37**

We thank the copyediting staff for their thorough treatment of our manuscript. All of the author queries have been addressed in the annotated document. We have also added additional references to make the manuscript as comprehensive and up to date as possible. All of the additions are merely to include the most accurate assessment of the literature; nothing about the general statements or conclusions of the manuscript are changed by these additions.